# A Critical Analysis of Indigenous Systems and Practices of Solid Waste Management in Rural Communities: The Case of Maseru in Lesotho

**DOI:** 10.3390/ijerph191811654

**Published:** 2022-09-15

**Authors:** Mpinane Flory Senekane, Agnes Makhene, Suzan Oelofse

**Affiliations:** 1Environmental Health Department, Faculty of Health Sciences, University of Johannesburg, Johannesburg 2028, South Africa; 2Nursing Department, Faculty of Health Sciences, University of Johannesburg, Johannesburg 2028, South Africa; 3SMART Places Cluster, Council for Scientific and Industrial Research, Pretoria 2000, South Africa

**Keywords:** environment, human wellbeing, indigenous practices, indigenous systems, rural areas, waste management

## Abstract

The aims of this study were to understand and to do a critical analysis of the different indigenous systems and practices of waste management to inform waste management policy development in Lesotho. To achieve these aims, the objective was to assess community perceptions of the impact of the indigenous systems and practices of solid waste management on the environment and human wellbeing. A simple random sampling method was employed. The primary data were collected through observations and survey questionnaires that were distributed among the communities in the study areas. The sample size was 693 participants from a total estimated population of 6917 in May 2021 in the Matsieng, Koro-Koro and Rothe constituencies. The data were analysed quantitatively by using the International Business Management Statistical Package for Social Science version 25.0. The descriptive method was used to interpret the results. For validity, the interview questions were set towards answering the study research questions. For reliability, open- and close-ended questions were designed. The research clearly indicated that indigenous systems and practices are culturally accepted in areas lacking formal waste collection services by the local authorities. The tradition, culture, values, and belief of the communities play a major role in the systems and practices implemented. Although some people convert waste items into useful products, the practices of general disposal were often unsafe from the human wellbeing and environmental perspectives. In Lesotho, a lack of awareness about conservation and sustainable use of natural resources could be attributed to flawed education at the grassroots level in schools.

## 1. Introduction

The Kingdom of Lesotho is a small, mountainous country that is surrounded by South Africa. The population stands at about 2.2 million people of which an estimated 60% live in rural areas. Maseru is the capital city with a population of about 120,000, of which an estimated 6917 reside in rural villages [1]. The World bank classifies Lesotho as a lower-middle-income country, with poverty levels having stagnated at around 30% between 2020 and 2021. Moreover, Lesotho has experienced unstable governments which have resulted in delays in development progress and meeting national goals. High HIV/AIDS prevalence and tuberculosis (TB) remain Lesotho’s greatest health challenges [2]. According to Alfthan et al. there was no sanitary landfill for Maseru before 2006 and waste was dumped into an old quarry and burnt, resulting in health hazards associated with air pollution [3]. Furthermore, toxic substances leached from the waste dumpsite into the city’s water reservoir. The UNDP initiated a project in Lesotho between 2009 and 2012 with the objective to support development of innovative public private partnerships for basic solid waste management service delivery. This project included waste collection, waste picking, street sweeping and recycling within the urban and peri-urban areas of Maseru [3].

However, waste management in Lesotho remains a challenge with indiscriminate dumping on roadsides, at markets and other public places being a common practice [4]. The [2] has estimated the 2016 waste generation rate in Lesotho at 0.11 kg/capita/day. The waste generation split by income category in Maseru is reported by the Bureau of Statistics [5] as 27.38 tonnes from low income, 41.98 tonnes from middle-income and 14.71 tonnes from high-income households. The absence of formal waste collection systems in the rural communities of Lesotho is a concern from both environmental and health perspective [1]. Uncollected household waste including soiled nappies, food waste, sand, gravel, paper, plastics packaging, metal, and glass are known to contribute to several environmental impacts [6] while also leading to human health impacts. The stagnation of water in waste items attracts mosquitoes and other insects, which breed and spread vector-borne diseases [6]. Food waste attracts flies, insects, rodents, and other vermin which also act as vectors that spread infectious diseases. Sepadi [7] reported that open burning of waste is a hazard to humans and the environment through air pollution and contributing to global warming [8]. Furthermore, there are incident reports confirming the impacts of waste on the local community. In 2015, in the Qoaling township, a toy from a dumpsite exploded injuring children, and in another incident in Lithabaneng township, medical waste mixed with commercial and industrial waste were found [9]. The impacts of solid waste management on the health of communities are not well understood and often perceived as an aesthetic problem rather than a public health problem but studies have confirmed public health risks throughout the waste cycle from generation to final disposal [10]. Furthermore, solid waste from industries and households poses a pollution threat to the drinking water source of Maseru, the Maqalika reservoir [11].

The knowledge gaps bridged by this study emphasize the connection between waste, the environment, health, and the Matsieng, Koro-Koro and Rothe constituencies’ (MKRC) communities. The gap in knowledge was identified to guide future policy development and improve health and wellbeing as it relates to individual rural areas. Seeman et al. [12] stated that the literature revealed gaps such as a lack of Waste Information Systems (WISs) in rural areas, a lack of health education on the proper management of waste in the rural areas, a lack of legislative frameworks that govern SWM in the rural areas, insufficient management of solid waste due to a lack of implementing existing laws and policies, and negative impacts on human wellbeing and the environment. Although some publications offer literature on the healthcare risks associated with waste management in urban areas, they do not provide solutions to help the MKRC communities overcome the barriers in their management of indigenous solid waste. Other publications offer SWM guidelines in the urban areas surrounding Maseru. In addition, Mothibe [11] conducted research focused on commercial facilities such as textile industries located at Ha Thetsane in the Maseru district. The Bureau of Statistics [5] similarly conducted a study on the weight of solid waste generated in the urban areas surrounding Maseru, and where human wellbeing and environmental degradation were elements of importance, studies were conducted on liquid waste. The researchers thus determined that no study has been conducted on indigenous systems and practices (ISPs) of SWM in the rural areas surrounding Maseru. Moreover, the rural communities on the outskirts of Maseru are not covered by Section 4 of the Environmental Act, No. 10 of 2008. This gap calls for rural communities to engage in ISPs of SWM. Therefore, there is a need for continuous research in rural communities looking at the negative impact of ISPs on human wellbeing and the environment. The potential to conduct further research focusing on causal relationships between waste, the environment and health in rural communities would inform policymakers to respond. Indigenous systems and practices (ISPs) of solid waste management (SWM) enable the rural communities of Maseru to collectively control their customary estates. These practices support the culture and promote the norms and values of rural communities. While new technological methods for waste management may be adopted to initiate smart ways to recycle and compost solid waste to divert it from uncontrolled waste disposal practices, such as illegal dumps and open burning. Gwimbi et al. [13] indicated that rural communities have a right to land, natural resources, and livelihood activities. One should understand that the rural communities collectively believe in their culture and want to maintain their sovereign rights and interests. McAllister [14] agreed that even though rural communities understand the environment and sustainable development, they are resistant to support new methods of waste management enshrined in waste management policies. Mphande [15] further built on the idea that rural communities have their own way of living and doing things in favour of their culture, norms, and values; it may take time for them to change their way of life. The aims of this study were to understand the different indigenous systems and practices of waste management, and to do a critical analysis of these systems and practices as evidence to inform waste management policy development in Lesotho.

In this article, we describe the methodology employed in this study and pay special attention to explaining the methodological approach and the data collection. We explain the quantitative descriptive design, ethnographic design, and the direct observation design. We further discuss data collection and data analysis. Next, we present the results of the study and provide the discussion where we focus on the perceived understanding of the community on the indigenous practices of SWM in the MKRC, the impact of indigenous solid waste on the environment, and the impact of indigenous solid waste on human wellbeing. This is followed by types of solid waste generated in the MKRC. Next is gaps in knowledge and then conclusions.

## 2. Methodology

This paper builds on the paper by [1] which outlined the methodological approach to investigate indigenous systems and practices relating to solid waste management in the rural areas surrounding Maseru. It was mandatory for this study to receive ethical clearance from the University of Johannesburg (UJ) and relevant authorities in Lesotho. Ethical clearance was issued by Research Ethics Committee at UJ. The ethical clearance number is National Health Research Ethics Committee Registration, REC 241112-035. Permission to enter the three constituencies (Matsieng, Koro-Koro and Rothe) in Maseru, Lesotho, was received from the Principal Chief of Matsieng (Chief Seeiso Bereng Seeiso) and the Principal Chieftainess of Rothe (Chieftainess Nthupi Bereng). The constituency of Koro-Koro was under the chieftainship of Chief Seeiso Bereng Seeiso. A letter from Chief Seeiso Bereng Seeiso was issued to the researcher to present to the headmen in the rural communities under study. Chieftainess Nthupi Bereng did not issue any letter, but through her secretary, informed all headmen under her chieftainess about the study. On arrival, the headmen confirmed that they were informed about the study. Appendix A is a clearance certificate from the University of Johannesburg; Appendix A is a letter of permission obtained from the principal chief of Matsieng in Maseru Lesotho. We present two maps of the study country and study areas.

Map of the study country, Lesotho.

Map illustrating some of the study areas indicated by arrows.



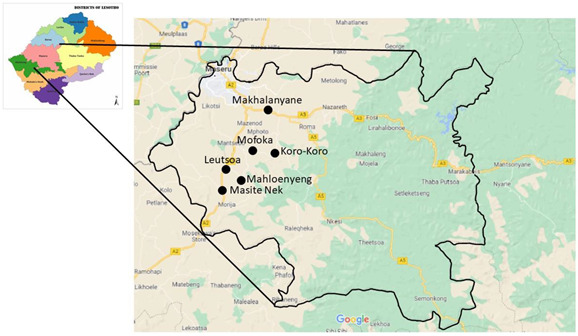



### 2.1. Explanation of Methodological Approach

We employed quantitative, qualitative and observations in this study. The primary data were collected with the assistance of four field workers and secondary data such as population estimates in the study areas were sourced from Lesotho government documents. From the internet, we accessed information on the description of the study country, the geographical location of the rural areas under investigation, and Lesotho demographics such as population, education, life expectancy at birth (Table 1), poverty rate, unemployment rate, types of solid waste generated, legislative frameworks and policies governing solid waste management in Lesotho. Quantitative descriptive data and observations were obtained without intervening in the participants’ daily activities. A qualitative ethnographic method was used to allow spending time talking to and observing people as they performed their daily activities in their natural settings. This approach assisted with developing a better understanding of their ISPs of SWM.

#### 2.1.1. Description of Data Collection Method

This study employed quantitative descriptive, ethnographic, and direct observational designs and they are discussed below.

##### Quantitative Descriptive Design

To answer the research questions about the real-life situation in the rural communities of Maseru, a quantitative descriptive technique was used [16]. The closed-ended questions included in the questionnaires were designed following a five-point Likert scale format to gather enough data in the research setting. A quantitative method was suitable to cover the various constituencies of rural communities of Maseru. Altogether, 693 respondents from the three constituencies in Maseru were randomly selected from a total estimated population of 6917 as obtained from Lesotho’s statistics office. An equal portion of 10% of the population in each rural community was considered to represent the total population and to maintain homogeneity and neutrality in the analysis and interpretation of the findings. Simple random sampling gives every member of the population an opportunity to be part of the study [17]. To conduct this method, a random number generator was used to select the population. Households were thus selected from rural villages within the three constituencies in the Maseru district.

##### Ethnographic Design

The qualitative ethnographic method was applied because researchers wanted to interact with respondents in their real-life environment. The interaction assisted to gain acceptance from the community so that they would freely respond to the questions without any hesitation. Furthermore, we aimed to generate knowledge about the behaviours, social structures, and shared beliefs of the rural communities of Maseru, Lesotho. Semi-structured interviews were conducted to obtain detailed information about the participants’ personal feelings, perceptions, and opinions. Participants were observed during their daily activities. Their activities were assessed and critically analysed to understand their indigenous practices and systems of SWM. This method was chosen to investigate the background of the research problem and collect qualitative data necessary for an explanatory analysis of the sample used for the study.

##### Direct Observation Design

Direct observation refers to a research design where the researcher collects data by observing what the participants do but also being mindful to not alter their environment or activities [18]. This is a qualitative approach to observing the community individually or in groups while in their natural environment [19]. Fieldworkers engaged in overt observation and informed the participants that they would be observed during their various waste management activities. Care was taken to observe from a sufficient distance to avoid interference with the waste management activities but to understand the ISPs of SWM in the rural communities of Maseru, Lesotho. Observations included all community members in Matsieng, Koro-Koro and Rothe constituencies (MKRC), who agreed to participate in this study after receiving permission from the principal chief and principal chieftainess. Observations were made before completion of the questionnaires if the research team found participants busy managing solid waste, or after the interview with the research team requesting participants to demonstrate how they managed solid waste. Observations were documented by means of taking notes and photographs of participants engaging in their daily SWM activities. This design assisted the researcher to bridge the gaps in understanding the research problem; connect with participants in their natural inhabited space; gather additional data that support literature on the topic; ask questions relevant to the specific cultural context; and realize new perspectives that take exception to the existing theoretical framework. The aim of the observations was to understand how the community engaged in the system and to understand the benefits of the indigenous system for rural communities.

### 2.2. Data Collection

Data were collected over a 40-day period between April and June 2021 and involved a questionnaire containing open- and close-ended questions, semi-structured interviews, and observations. Interviews were conducted during the week from Mondays to Fridays between 07:00 and 17:00 daily. Two additional focus group meetings involving ten (5 male and 5 female) former employees of the Maseru City Council (MCC) with the knowledge of waste management from a government perspective. These meetings were called to fill the knowledge gap when authorities in the Ministry of Environment in Lesotho declined to participate. The former employees were recruited through WhatsApp invitations and the meeting was held at a local high school. We used the quantitative method to collect numerical data, and an Excel spreadsheet was used to record the transferred raw data obtained from the survey questions. Qualitative findings were expressed in words, and this approach was chosen to allow the researchers to interview participants one-on-one using semi-structured open-ended questions. Observations were chosen to observe participants’ behaviour in terms of their practices. Semi-structured interviews were chosen because it was necessary for the researchers to prepare the questions. Researcher should prepare questions that are relevant to the study, that will be understood by respondents and that will not consume time for both respondents and the researcher. Using these types of interview questions provided us with an opportunity to structure the questions in the format we preferred. The researcher worked towards a flexible structure and reliable qualitative data; therefore, the use of semi-structured open-ended interviews was an advantage. This study was conducted in Lesotho, and because of COVID-19 regulations, the researchers had to complete the data collection. To avoid multiple interview rounds, semi-structured interviews were conducted to gather detailed information using twelve questions. In this study, these types of questions were meant to subjectively analyse participants’ opinions in terms of how they understand indigenous SWM; seek participants’ opinions on translating useful ISPs into effective SWM; and understand participants’ opinions.

### 2.3. Data Analysis

For quantitative analysis, we checked for missing data, outliers were removed, and raw data were transformed into significant and understandable information using the following three steps: data validation, data editing, and data coding. Data were analysed in terms of ISPs of SWM. To communicate the results, a percentage was used to determine the number of respondents. For qualitative analysis, we focused on language, pictures, and observation. One-on-one interviews were transcribed, and a thematic analysis method was employed. Data were coded before identifying and reviewing the themes that were examined separately to gain an understanding of participants’ perceptions of ISPs of SWM and their motivations. We used the software International Business Management Statistical Package for Social Science version 25.0.

## 3. Results and Discussion

The number of respondents indicated as the sample size relative to the population in the study area is indicated in Table 2 and the demographics of the respondents are summarised in Table 3. 

### 3.1. Community’s Perceived Understanding of Indigenous Practices of SWM in the MKRC

For this section, the number of responses varied for each question as participants did not always answer all the questions. The first question (E1.1) with 463 responses asked whether waste is stored and the purpose of the storage. Six scenarios were provided for respondents to indicate their understanding on a Likert scale (Figure 1). Out of these, 50.5% strongly agreed, 15.3% agreed, 3.2% were neutral, 22.7% disagreed, while 8.2% strongly disagreed. The range of responses indicated that storage practices differ among individual community members and depended on the type of waste that is stored. For instance, Liqo (maize stalk) is stored in large sacks, while other kitchen waste is stored in buckets until it is fed to dogs and pigs. The second question (E1.2 in Figure 1) explored the storing of dry cow dung to use as fuel; responses were received from 464 participants. Over 95% of respondents (90.7% strongly agreed, 4.5% agreed) agreed that dry cow dung was stored to be used as fuel. The next question (E1.3, Figure 1) tested if cow dung was used to run a generator and over 95% (15.1% disagreed, 83.8% strongly disagreed) of respondents disagreed with this statement. Testing if cow dung was used to provide lighting in their homes (E1.4, Figure 1) revealed that 97.8% of respondents did not use cow dung for lighting purposes (19.4% disagreed, 78.7% strongly disagreed). Question E1.5 Figure 1) asked whether the MKRC communities store waste that was capable of decomposing through bacteria to use as garden manure. More than 86% (76.4% strongly agreed, 10.2% agreed) of the respondents agreed with statement, while 10.6% (0.6% disagreed, 10% strongly disagreed) did not agree with statement. The last question in this category asked. whether they stored waste capable of decomposing through bacteria to produce new items, 62.9% strongly agreed, 15.5% agreed, 8.6% were neutral, 5.8% disagreed, and 7.1% strongly disagreed. It is therefore quite clear that most community members stored waste for different purposes, but cow dung was mostly stored to be used as a direct fuel source.

E1.1 explored people’s perception of the impact of indigenous SWM on the environment. The analysis asserts that the majority agreed or strongly agreed that they understand waste must be stored properly in buckets, bags, and other containers before its disposal. It is concluded that buckets and bags used to store waste before disposal are ideal for containing waste. These containers store waste that is intended to be used in the future; participants claimed they protect the waste from rain which can cause contaminated runoff. From the minority of participants who disagreed or strongly disagreed, it can be interpreted that they had limited knowledge of the usefulness of waste and how it can contaminate the environment if not properly managed. E1.2 stated, “People in this community store dry cow dung to use as fuel”. Responses confirm participants were from typical rural areas where people engage in farming and raise domestic animals such as cows, horses, sheep, and goats. A majority depended on their herd of cattle for sources of fuel. The minority could be members of the communities without domestic animals. Most of those who used cow dung for fuel lacked knowledge of the health implications of the smoke generated when cow dung is burnt, indicating they are susceptible to respiratory diseases. E1.3 sought responses to: “People in this community use cow dung to run a generator”. This analysis asserted that the rural communities use cow dung as a valuable source for cooking and heating their homes only, and not for any other purposes. It also means that they rely more on traditional resources than modern resources. E1.4 concerns itself with “People in this community store dry cow dung to light up their houses”. This analysis suggests that raw cow dung was not an ideal source of renewable energy. Alternative approaches, such as training the communities to convert cow dung into electricity, can benefit the MKRC communities. E1.5 stated: “People in this community store waste that is capable of being decomposed by bacteria to use as garden manure”. Most participants agreed or strongly agreed. It is evident that most participants understood waste cannot be converted into manure without the biological activity of microorganisms such as bacteria. This means they can distinguish between the types of waste capable of decomposing and that such waste cannot be disposed of anywhere; it can contaminate the environment if not disposed of safely. E1.6 relates to: “People in this community store waste that is not capable of being decomposed by bacteria to produce new items”. The analysis suggested that most community members did not consider waste as something that is unwanted but as an item that can be reused or converted to make new items. Conversely, the analysis asserted that the minority of participants who disagreed or strongly disagreed with this statement might be individualists who were conscious of the negative effects of rodent and cockroach infestations caused by the long-term storage of non-degradable waste. They would not like to have rodents and cockroaches in their homes as they transmit diseases to them and their families.

No measurements, such as the weighing of waste, were made in the study areas because of a lack of equipment. Percentages obtained from government of Lesotho documents [5] are comparable to waste generated in the urban areas of Maseru and summarised in Table 4. It is obvious from these documents that paper waste is predominant (43%) in the study areas, with a very low percentage of glass waste at 2%. It is not surprising, however, that kitchen waste is generated at 19%. This study contends that the cause may be that some of the community members do not have access to electricity and, therefore, may not have refrigerators to store their leftover foodstuffs. Plastics (20%), Paper (43%), Glass (2%), Kitchen leftovers (19%), garden refuse (4%), others 3%. Garden refuse is sometimes used to feed domestic animals such as pigs, goats, sheep, and cows; hence, there is 9% waste generation. Moreover, while it has become the norm that plastics are used as carrier bags for groceries, in the case of the study areas, plastic waste may be low because many households are not working and do not visit grocery stores on a regular basis. Another reason is that some households collect plastic to produce new items, such as plastic mats, handbags, and hats.

There is an understanding amongst the community members in the study areas that culture does not support the disposal of waste on the ground. This is evident from the responses of a majority who disagreed or strongly disagreed with this statement. This analysis asserted that for some members of the community, culture may be disappearing, so it is reasonable to encourage communities not to ignore their cultural beliefs and waste management practices because cultural beliefs have an influence on present waste disposal methods. In addition, cultural beliefs may contribute positively to alleviating indiscriminate waste disposal problems. Not everyone lived in proximity to natural resources. This analysis asserted that there were natural resources in the MKRC ancestral territories, but these were somehow exploited. It is clear then that there is a need to enhance environmental sustainability through interdisciplinary and multidimensional approaches such as cultural constructs, Furthermore, MKRC communities were not bonding with their natural resources, or they had limited natural resources that were accessible to a certain portion of community members. This analysis concluded that it is important to illustrate effective systems and practices of SWM through human interconnectedness, the environment, and sustainable development. Most community members still followed the footsteps of their ancestors when it came to waste management. It shows that most community members understand their culture, thus they continue observing their rituals.

There were 464 respondents to the questions interrogating the culture and social systems in the study area. In response to the question if indigenous systems of managing solid waste in their area could be attributed to peoples’ cultural beliefs 62.5% agreed (32.3% strongly agreed, 30.2% agreed) while 24.6% disagreed (16.2% disagreed, 8.4% strongly disagreed) (D1.1 in Figure 2). On question D1.2 asking whether indigenous systems of managing waste in their area could be attributed to people’s cultural values, 60% of respondents agreed (32.4% strongly agreed, 27.6% agreed) and 24.6% disagreed (16.2% disagreed, 8.4% strongly disagreed). Assessing whether it was out of place in their culture to dispose of waste on the ground (D1.3) 92.4% (17.0% disagreed, 75.4% strongly disagreed) disagreed with the statement (Figure 2), clearly indicating that waste disposal on the ground is acceptable from a cultural perspective. When asked whether people live in proximity to natural resources and tend to employ a WM system that governs natural resource use (D1.4) 57.4% agreed (33.7% strongly agreed, 23.7% agreed) while 31.7% disagreed (7.8% disagreed, 23.9% strongly disagreed) and 10.9% responded neutral to this question. These results indicate a weak collectiveness that holds communities in the MKRC together in their cultural beliefs. This may have a negative impact on individual waste management practices. The analysis also suggests that cultural values in the study area are focused on personal values and have failed to consider value dimensions within a broader framework of the community. Consequently, there is no relationship between the values and behaviour of individual members of the MKRC communities, indicating that communities in the MKRC are driven by individualistic cultural values. It is therefore reasonable to conclude that individuals’ values may influence their level of interest in waste management and can affect the systems and practices of SWM in the study area. Participants were asked if they believed a natural environment could be successfully conserved using peoples’ cultural mores (D1.5, Figure 2). The responses were 47% in agreement (25.2% strongly agreed, 21.8% agreed) 17.5% neutral, and 35.6% did not agree (15.3% disagreed, 20.3% strongly disagreed). This means MKRC communities were not bonding with their natural resources, or they had limited natural resources that were accessible to a certain portion of community members. This analysis concludes that it is important to illustrate effective systems and practices of SWM through human interconnectedness, the environment, and sustainable development. On the question of whether the indigenous systems of WM that people practice are based on people’s customs, rituals, and traditions (D1.6 in Figure 2), 463 participants responded of which 88.5% agreed (55.5% strongly agreed, 33.0% agreed) and only 3.7% disagreed (2.6% disagreed, 1.1% strongly disagreed). Responses to this question indicated that most community members still followed the footsteps of their ancestors when it came to waste management. It shows that most community members understand their culture, thus they continue observing rituals.

Participants in the study areas indicated that they used waste items such as old tyres to make chairs (Figure 3), plastic to make plastic hats, and animal skin to make Basotho blankets. There is consistency in these systems between the findings from the data collected and literature. McCombes [21] referred to Namibia and stated that glass, paper, metal, plastic and electronic waste items were recycled. Communities in the study areas, similarly, collect recyclable waste items such as plastics, scrap metals, glass, and cardboard and sell them to recycling companies. The difference that this study identified is that communities in Namibia used technology to convert waste items, while communities in the study areas employed indigenous systems in some waste items to showcase their intellectual capabilities in craft.

### 3.2. Community’s Perceptions of the Impact of Indigenous Solid Waste on the Environment

When asked if waste disposal practices cause damage to the environment (F1.1, Figure 4), 464 participants responded of which 60.3% agreed (37.7% strongly agreed, 22.6% agreed), 8.4% were neutral, and 31.2% disagreed (14.2% disagreed, 17% strongly disagreed). The support for this statement illustrated a correlation between waste disposal practices and knowledge of the impacts that disposal may cause to the environment. Further interrogation on whether the negative impacts included serious risks such as the transfer of pollution to ground, water, and land (F1.2, Figure 4) solicited 462 responses, of which 67.5% agreed (59.3% strongly agreed, 8.2% agreed), 18.8% remained neutral in their response, and 13.6% disagreed (8.4% disagreed, 5.2% strongly disagreed). The analysis suggested that the respondents who agreed with this statement may be living in areas where they could detect a smell with no obvious origin, or they have observed changes in the landscape that they could associate with waste disposal. Another question was whether there was a need to address an understanding of the indigenous management of solid waste for rural communities of Maseru through concerted efforts to improve indigenous waste management (F1.3, Figure 4). There were 464 responses, of which 98.1% agreed (96.8% strongly agreed, 1.3% agreed), while only 1.5% disagreed (0.4% disagreed, 1.1% strongly disagreed). This is a clear indication that communities need the help of various stakeholders such as Maseru local government officials, waste experts, solid waste private sectors and academics to improve the current indigenous solid waste systems in the respective areas. When participants were asked if waste activities generated GHG (F1.4, Figure 4), only 11% of 461 respondents agreed, while 14.8% responded neutral, and 74.2% disagreed (7.8% disagreed, 66.4% strongly disagreed). This question is specific to the study area. The analysis asserted that participants knew how GHG is emitted into the atmosphere. The answers presented earlier identified cow dung as a source of fuel used for cooking and heating in the homes; in this way, cow dung alone may not be used as a qualifying source for generating GHG. Other sources of GHG such as manufacturing, transportation, electricity, and deforestation in the study area also contribute to GHG emissions. The last question was about leachate that comes from informal dumpsites contaminating the surface and groundwater (F1.5, Figure 4). There were 461 respondents of which 51.7% agreed (44.5% strongly agreed, 7.2% agreed), 21.9% remained neutral, and 26.5% disagreed (10.2% disagreed, 16.3% strongly disagreed). The analysis suggested that it is highly improbable that leachate can be formed in the study areas because people do not generate much waste because of the high unemployment rate. The biodegradable waste that is generated is stored in containers and allowed to decompose so that it can be used in the future as garden fertilizers. Those who did not store their waste burned it, and the ash was disposed of in a specific area, either at the corner of individuals’ yards or in a communal area (Thotobolo) far from residential homes.

### 3.3. Community’s Perceptions of the Impact of Indigenous Solid Waste on Human Wellbeing

The first question under this sub-category was whether a lack of knowledge of the consequences of poor SWM had increased the occurrence of infectious diseases. There were 463 responses, and 75.6% strongly agreed, 11.4% agreed, 4.1% were neutral, 1.7% disagreed, while 7.1% strongly disagreed. On the question of whether infectious diseases include those that are caused by various vectors such as rodents and mosquitoes, there were 464 responses, and 76.5% strongly agreed, 13.4% agreed, 9.3% were neutral, 0.6% disagreed, and 0.2% strongly disagreed. When participants were asked whether offensive odours from heaps of solid waste affect human beings, 464 responded, and 88.89% strongly agreed, 10.11% agreed, 0.6% were neutral, 0.2% disagreed, and 0.2% strongly disagreed. The last question was whether human health problems are caused by ignorant waste generators. There were 464 responses, 74.4% of which strongly agreed, 16.2% agreed, 7.3% were neutral, 0.2% disagreed, and 1.9% strongly disagreed. Figure 5 below shows how respondents perceive the impact of indigenous systems of waste management of human wellbeing. 

Lack of knowledge on the consequences of poor SWM has increased the occurrence of infectious diseases. This study revealed that communities generate infectious waste such as used nappies and waste from slaughtering animals. Infectious waste contributes to infectious disease through environmental and water contaminants as well as rats and mosquito infestations. Communities in the study areas are denied access to municipal services, some of which include health education on how best to handle solid waste to prevent the occurrence of infectious diseases. Infectious diseases include those that are caused by various vectors such as rodents and mosquitoes. When waste is disposed of improperly and remains there for a long time, the result is rodent and mosquito infestations. Rodents and mosquitoes transmit infectious diseases if not controlled or prevented. We suggest that many communities keep waste in one place for a long time, causing infectious diseases. As illustrated in Figure 6 below, Offensive odours from heaps of indigenous disposal of solid waste affect human beings. This implies that waste is disposed of without being treated to render it free from offensive odours, which ultimately affects human wellbeing. In-depth interviews also revealed that people react differently to odours because they are not equally sensitive to chemicals found in the waste and therefore may not be affected in the same way. However, we conclude that one cannot rely on odours to determine the level of health risks posed by waste. Instead, it is suggested that when odours caused by decomposed waste persist, doors and windows can be closed until a solution to controlling and preventing such odours is considered. Ultimately, participants’ waste generation is linked to employment status and is considered an individual problem, while waste management could be attributed to communities’ cultural beliefs. We suggest that waste should be handled and disposed of in a satisfactory way that does not cause a nuisance. Human health problems are the origins of ignorant waste generators. Ignorance goes along with human behaviour. It is evident that some human health problems originate from ignorant behaviour; therefore, it is imperative that people are educated about waste management, disposal and health problems associated with solid waste practices.

### 3.4. Types of Solid Waste Generated in the MKRC

Amongst the three analysed constituencies, Rothe was found to be the only one that did not have evidence of stored corrugated iron sheets, tyres, cow-dung and garden waste and the questions are: What do they do with these types of waste that they have generated? Is there a common place where they dispose of these types of waste that was not observed by the researcher during data collection? The most common heavy metals linked with the types of waste generated were from the highest to lowest: Lead (Pb) was found to be associated with a few wastes generated, followed by cadmium (Cd) and then copper (Cu) and zinc (Zn) with equal levels of concentration. Chromium (Cr) was followed by arsenic (As). Barium (Ba) and nickel (Ni) have equal levels of concentration in solid waste, then magnesium (Mg). The last four heavy metals with equal levels of concentration were mercury (Hg), silver (Ag), iron (Fe) and cobalt (Co). Table 5 lists the different types of waste generated in the MKRC.

Table 5 lists the types of solid waste generated in the rural areas of Matsieng, Koro-Koro and Rothe. The Matsieng and Koro-Koro constituencies have similar types of generated solid waste, while Rothe differs from them because it did not generate corrugated iron sheet, cow dung and garden waste. 

We did not find literature that links corn stalk with any of the heavy metals mentioned in this article. Corrugated iron sheet contains Cu, As and Zn [23]. Dorenfeld et al. [24] stated that plastic contains Cd, Zn, lead (Pb) while Senekane [20] showed that tin contains Pb, Zn, Cr, Ni, Cu, As and Cd. Aravind, Sharath, Reddy stated that Fly Ash contains Pb, Hg. Glass contains Pb, Cd, Cr and Ba [25]. Oyen et al. [26] stated that fabric contain Ba. Authors like [27] postulated that tyres contain petroleum hydrocarbons (PHC), As, Pb, Cr, Ba. According to [28], wood contains heavy metals such as Cd, Pb, Cu, Zn, Ni, Mn. Duda et al. [29] stated that paper contains heavy metals such as Cd, Zn, Pb, Cu, Cr, Ni. Solid waste like steel contains Pb, Cd, Hg, Ag, As [30]. Elmas and Cinar showed that cow dung contains Fe, Cu, Mn, Ni Zn, Cr, Pb, Co, Cd [31], while Emgwa et al. [32] stated that garden waste contains Pb, Cd, Cu, Zn. Singh et al. [33] indicated that heavy metals present in leachates from both hazardous waste dumps and municipal solid waste landfills pose a serious threat to public health, because they can cause several physiological effects to human health. According to [34] lead is a health hazard because if ingested through contaminated food or drinking water, it affects the soft tissues and skeletal bones. Older homes painted with lead-based paint are other major exposure pathways. Considering this, it is imperative that environmental health practitioners in Lesotho play a major role in educating rural communities about health hazards of heavy metals. Jaishankar et al. [35] indicated that “due to their toxicity, non-biodegradability and persistency, heavy metals can exert adverse effects on the environment and other ecological receptors. Therefore, their removal from soil and aqueous environments has drawn tremendous attention. Various methods have been developed and used to decrease heavy metals concentrations in the ecosystems. These technologies can be categorized in physico-chemical processes such as ion exchange, reverse osmosis, membrane filtration, adsorption, precipitation, electrolytic removal, and biological processes involving activated sludge and phytoremediation”.

Waste containing these heavy metals can contaminate water sources if disposed of in the landfill site; hence, there is a need to avoid landfill disposal of solid waste. It is clear therefore, that water bodies could be contaminated by the metals found in leachate. It is concluded that in the rural areas like MKRC where most community members are unemployed, there is small amounts of waste biodegradable waste generated, the leaching behaviour may not be easy to determine. It was observed during data collection that the type of household waste that is generated in the MKRC is not disposed of into the environment, but it is used to feed animals such as pigs, dogs, and cows. Pigs and dogs are fed from kitchen food waste while cows are fed from garden waste. Papers/card box are used to make fire for cooking; plastics are reused to neat hats, mats, and handbags. This explains that non-biodegradable waste in the rural areas of MKRC is not disposed of but used for various purposes. The study’s findings reflected 12 themes related to ISPs of SWM, and these are discussed as follows: 

Theme 1: Understanding ISPs of SWM—The findings show that, in general, people understand ISPs of SWM from cultural, norms and customs perspectives. People in their respective areas use various pathways to manage their waste including the implementation of a waste hierarchy (reusing and recycling) and local management systems and practices. This is consistent with what [36] wrote in their publication; they postulated that indigenous communities around the globe use various pathways to manage their indigenous solid waste and gave the following examples: Territorial management practices and customary governance, contributing to nature conservation and restoration efforts with regional global implications, countering the drivers of unsustainable resource use, and resisting environmental injustices. This section concludes that ISPs of SWM differ based on culture and traditional beliefs, customs, values, and geographical location.

Theme 2: Recognising ISPs of SWM in addressing local environmental issues—the findings reveal that in the study areas, there were different opinions, potentially attributed to individual levels of education and lack of understanding and knowledge of the impact caused by improper handling and disposal of solid waste. Participants indicated that some community members recognise ISPs of SWM in addressing local environmental issues, while this is not the case for others. However, this is inconsistent with what [37] found in Canada, and they postulate that waste disposal practices were similar among all community members. In support, [38] proclaimed that the burning of waste (in particular plastics) has become dangerous to the environment and human wellbeing despite the guidance from authorities to engage in best practices of SWM.

Theme 3: Enhancing communal waste management—this study found that it is a common practice for many rural communities to engage in unsafe SWM practices. Common practices include digging holes in the backyards of homes, burning the waste, or disposing of waste in the streets. Participants indicated that each member of the community manages waste in the way most suitable for them unless the headmen instruct community members to engage in cleaning campaigns. Suitable methods include the collection of recyclable items such as bottles and aluminium cans, which are then sold to a recycling company at Ha Mantsebo in the outskirts of Maseru City. This is consistent with literature, where [39] referred to Nanjing in China, indicating that SWM is enhanced through waste recycling. Chen et al. [39] postulated that the informal waste pickers receive support from recycling companies and the government through the implementation of relevant laws, regulations, and policies. A noticeable difference between Nanjing and this study’s findings is that, in the study areas, the collection of recyclables is done informally without consideration of laws, regulations, and policies. Dururu et al. [40] researched Northampton, Milton Keynes, and Luton in the East Midlands of England, and found that community members engaged in various activities to manage their waste. One example was SWM awareness campaigns. Moreover, unwanted clothes are not disposed of as waste but are given to the needy as donations. The literature is thus consistent with this study’s findings because the environment is kept clean from pollution by avoiding the illegal dumping of solid waste.

Theme 4: Preventing the breeding of mosquitoes and other health problems—the findings show that where there were no municipal solid waste collection services, people dispose of the waste anywhere they find a space to do so and, in many cases, this becomes a source of mosquitoes, which then transmit diseases. The participants claimed that decomposed waste is a source of mosquito breeding, which causes certain health problems in humans. This is consistent with previous studies conducted by [41] who found that the spread of disease resulted from blocked waterways causing mosquitoes to breed in waste canals, and illegal dumpsites were posing a risk of spreading diseases. Kumar et al. [10] confirmed this by indicating that the uncontrolled disposal of solid waste in stagnant water is a source of mosquitoes, which then transmit diseases to humans. The difference between the findings and literature is that in the study areas, participants engaged in illegal dumping because they did not have municipal waste collection services. The findings from the literature illustrate certain rural communities engage in illegal dumping because there are no municipal services or services are inconsistent, so when significant waste is generated, community members do not wait for waste collection but settle on illegal dumping practices.

Theme 5: Preventing bad smells from decomposed waste—a lack of municipal waste collection services results in bad smells emanating from decomposed waste. The participants confirmed there were no municipal solid waste removal services in their areas. A lack of municipal services accounts for unpleasant odours. This is consistent with what [42] stated in his study conducted in Nigeria on indigenous systems of SWM. Yakabu [42] found that indigenous solid waste practices are an eyesore and produce unpleasant odours. Authors [6] and [41] supported the statement by [42] and postulate that waste dumps cause bad odours. Mihai and Taherzadeh [43] similarly mentioned the odours caused by heaps of waste that is improperly managed, and this study confirms the same phenomenon.

Theme 6: Converting waste into useful items—various items are recycled to produce new products. In other parts of the world, cast-offs are donated to destitute communities. The participants in the study areas indicated that they used waste items such as old tyres to convert them into chairs, plastic into hats, and animal skins into Basotho blankets, among others. There is consistency in these systems between the findings and observations in literature. Dorenfeld et al. [24] claimed in Namibia, glass, paper, metal, plastic, and electronic waste items were recycled. Communities in the study areas similarly collected recyclable waste items such as plastics, scrap metals, glass, and cardboard and sold these to recycling companies. The difference this study identified is that communities in Namibia used technology to use the waste items while communities in the study areas employed indigenous systems to showcase their intellectual crafting capabilities.

Theme 7: Managing kitchen waste—in some African countries like Egypt, communities feed kitchen waste to animals, such as pigs and dogs. In some European countries, like Sweden, kitchen waste is disposed of in drains. The participants mentioned that waste from the kitchen is used to feed domestic animals such as pigs and dogs. In support, Azmat [44] indicated that Egyptians feed pigs on waste generated from their kitchen. However, in contrast, Azmat [44] also reported that a small percentage (8%) of kitchen waste is disposed of in compost plants and the rest in open spaces. There are thus differences in the kitchen waste practices between study findings and literature. According to Dikole and Letswenyo, food waste in Botswana occupies a high percentage (of unspecified figures) of moisture, but it is unclear how communities in this country solve kitchen waste challenges [45].

Theme 8: Translating useful ISPs into effective SWM—molora (ash) is mostly used for pest control in gardens and farms. In the study area, it is used for various purposes, including treating unknown diseases, as a symbol for death in the house, and others. To ensure sustainable vegetable production, waste items such as ash, bones, cans, and animal dung are added to what is referred to as “Lentloane”. This practice is employed in the study areas and by international organisations such as food and agricultural organisation’s UKaid and USAID. The participants reported that they had several ways in which they translated useful ISPs into effective SWM. The findings were based on both quantitative and qualitative methods of data collection on uses of molora (Ash), and they are as follows: As one of the ingredients to manufacture traditional paint for their homes; to destroy mafokololi (caterpillars) in the gardens or fields; applied to windows to symbolise a death in the house; applied to the human body to treat unknown skin diseases; it is spread on the skin of newly slaughtered animals to prevent flies. The thotobolo (ash heap) is used to accommodate a copper pole with forks to prevent lightning strikes. There is consistency in the use of molora for home gardens and agricultural farms between this study’s findings and literature. Ramraj and Ramsingh [46] explained that for Trinidad and Tobago Islands, close to South America, communities use wood ash to control pests in their home gardens. McCoid and Hainey [47] referred to the United Kingdom and postulated that wood ash is used to control ticks in the home garden. In addition, Ndlovu and Sprickerhoff [48] referred to Zimbabwe and Ghana and indicated that they use wood ash to protect maize from pest attacks by mixing maize kernels with wood ash. Bharathi et al. [49] stated that in India, ash produced from the stem juice of *Musa paradisiaca* Linn is valuable in healing wounds. The difference identified here is that this study’s participants explained ash is broadly used in treating diseases, while the literature confirmed the use of ash is specific to wound treatment. This study concludes that ash is an important resource for preventing and destroying garden and farm pests in certain countries across the globe. It is also a useful resource for treating unknown diseases. 

Theme 9: The disappearance of the ISPs of SWM—considering that there are no municipal waste collection services in all geographical areas, ISPs of solid waste will never disappear. The findings showed that ISPs in the MKRC will never disappear if there are no municipal waste collection services. Besides, the ISPs of SWM are culturally accepted. This is consistent with literature [12] on rural and remote first nations in Canada, where communities consistently dispose of their generated waste through burial methods and in open dumps because there are no municipal waste collection services in their areas. In Ghana, Kosoe et al. [50] also found that community members supported indigenous methods of waste management because of their cultural beliefs. In addition, communities knew how to manage their waste properly using ISPs. However, officials responsible for waste management in Ghana did not support the indigenous methods but were in favour of new technologies of SWM. Moreover, private sectors were also supported over public sectors to take control and employ new technological methods of waste management in this country. One noticeably important factor is that culture plays a significant role in indigenous waste management. In addition, rural communities with no municipal services still value the beauty of their environment and health even though they have no choice but to engage in indiscriminate solid waste disposal methods.

Theme 10: Relevance of the ISPs of SWM for the conservation and sustainable use of natural resources—ISPs of SWM are relevant to conservation and the sustainable use of natural resources, but education on the topic at the grassroots level is flawed in schools. The findings showed that ISPs in the MKRC will never disappear if there are no municipal waste collection services. Besides, the ISPs of SWM are culturally accepted. This is consistent with literature [12] on rural and remote first nations in Canada, where communities consistently dispose of their generated waste through burial methods and in open dumps because there are no municipal waste collection services in their areas. In Ghana, Kosoe et al. [50] also found that community members supported indigenous methods of waste management because of their cultural beliefs. In addition, communities knew how to manage their waste properly using ISPs. However, officials responsible for waste management in Ghana did not support the indigenous methods but were in favour of new technologies of SWM. Moreover, private sectors were also supported over public sectors to take control and employ new technological methods of waste management in this country. One noticeably important factor is that culture plays a significant role in indigenous waste management. In addition, rural communities with no municipal services still value the beauty of their environment and health even though they have no choice but to engage in indiscriminate solid waste disposal methods.

Theme 11: Facilitation of cheap, effective, and sustainable community environmental cleanliness—clean-up campaigns are cheap and can be participated in by everyone, young and old, men and women, from any private or public sector. They can be organised and implemented at any time during the year. The report [51] stated that “A clean and safe environment and healthy residents are the ultimate goals of environmental justice, smart growth, and equitable development”. It is not uncommon for rural communities on a global scale to face an array of challenges associated with clean and safe environments. In Lesotho, cleaning campaigns are focused on urban areas only and neglect the rural areas. This is confirmed by [52] who referred to a cleaning campaign that was organised by the prime minister of Lesotho, Dr. Moeketsi Majoro. The cleaning campaign took place in Maseru City in September 2020. In addition to this, Kajane [53] referred to a cleaning campaign in which the World Environmental Day was marked in Maseru City, where participants included Vodacom Lesotho, the United Nations Lesotho, British High Commission in Maseru, and the American International school; these organisations hosted a clean-up campaign in their respective jurisdictions. Furthermore, Ramatlapeng [54] referred to a national cleaning campaign that took place in Maseru City on 25 November 2020. For this campaign, the MCC received donations of 100 refuse bins from the Central Bank of Lesotho. Participants emphasised that their headmen seldom organised cleaning campaigns. This is consistent with what is reported in the previous paragraph [52,53,54]. The difference is that it is apparent there is an organized cleaning campaign that takes place each year in the capital city, Maseru, where participants and organisers are in high-ranking positions at the national level and government entities. This is not the case in the study areas where the headmen act as organisers and governments are non-partisans. In the study areas, cleaning campaigns only take place after unspecified periods. In addition, cleaning campaigns in the study areas are questionable because there is no documented evidence of what participants reported. Furthermore, cleaning campaigns that took place in Maseru City in 2020 and 2021, where the prime minister did not participate, are also dubious because there is no documented evidence like photos. Appendix A is an illustration of an organised cleaning campaign in Maseru.

Theme 12: Exhibition of cultural beliefs and values through the practice of the indigenous systems of SWM—the participants expressed their understanding of ISPs of SWM across all age groups. The participants referred to cultural mores, traditions, and practices as important parameters throughout the waste management process in their respective areas. It is evident from the findings of this study that communities in the study areas listen to their authorities (headmen) when instructed to participate in a cleaning campaign in their respective areas. This study’s emphasis on cultural mores is consistent with what transpired in the literature, where studies mentioned ISPs of SWM. Panta [55] referred to the importance of cultural mores and indicated that culture in developing countries is based on oral communication, where communities practically engage in SWM activities after receiving instructions from the authorities. This study finds it important for each community member to show a sense of socialisation and conform to the norms of their culture. Showing respect to their authorities’ instructions would mean that they understand they should share the rules and expectations of their culture and behave appropriately. Roberts et al. [56] mentioned how rural communities manufacture manure/fertiliser for their crops and postulate that, based on their culture, communities in the western part of Nigeria left organic waste to decompose so that it could be used as manure to grow crops. Such wastes include food waste, animal faeces and dead plants. It is also concluded that it is crucial for communities to achieve a sense of belonging; thus, they will be safe and keep abreast of the standards in their respective areas. This study considers values to be a specific cultural standard, and this is where communities need to share collective values. Obasiohia [57] similarly referred to Nigerians in rural areas who use their culture to manage solid waste. A noticeable difference in the findings and the existing literature is where Ngara [58] referred to rural communities that live close to their natural resources and believe they can conserve their natural environment through their tradition, cultural mores, and practices. This study asserts that there is a weak collectiveness in the MKRC that holds communities to their cultural beliefs. Instead, communities focus on personal values, and this tends to isolate certain members of the communities from the group. As such, it is reasonable to confirm that there is a barrier in culture because the findings make it clear that individualist cultural values influence how systems and practices of SWM are conducted in the study areas.

## 4. Gaps in Knowledge

The knowledge gaps bridged by this study emphasize the connection between waste, the environment, health, and the MKRC communities. The gap in knowledge was identified to guide future policy development and improve health and wellbeing as it relates to individual rural areas. Gaps identified during the data collection in the MKRC include a lack of Waste Information Systems, a lack of health education on the proper management of waste, a lack of legislative frameworks that govern SWM, insufficient management of solid waste due to a lack of implementing existing laws and policies, negative impacts on human wellbeing and the environment. There were no previous studies found which could be used to provide solutions to help the MKRC communities overcome the barriers in their management of indigenous solid waste. Moreover, the rural communities on the outskirts of Maseru are not covered by Section 4 of the Environmental Act, No. 10 of 2008. This gap calls for rural communities to engage in ISPs of SWM. Therefore, there is a need for continuous research in rural communities looking at the negative impact of ISPs on human wellbeing and the environment. The potential to conduct further research focusing on causal relationships between waste, the environment and health in rural communities would inform policymakers response.

## 5. Conclusions

The study’s findings reflected 12 themes related to ISPs of SWM, and these are summarized below. Theme 1: Understanding ISPs of SWM—the findings show that, in general, people understand ISPs of SWM from cultural, norms and customs perspectives. People in their respective areas use various pathways to manage their waste. Theme 2: Recognizing ISPs of SWM in addressing local environmental issues—the findings reveal that in the study areas, there were different opinions, potentially attributed to individuals’ level of education and lack of understanding and knowledge of the impact caused by improper handling and disposal of solid waste. Theme 3: Enhancing communal waste management—this study found that it is a common practice for many rural communities to engage in unsafe SWM practices. Common practices include digging holes in the backyards of homes, burning the waste or disposing of waste in the streets. Theme 4: Preventing the breeding of mosquitoes and other health problems—the findings show that where there were no municipal solid waste collection services, people dispose of the waste anywhere they find a space to do so and, in many cases, this becomes a source of mosquitoes, which then transmit diseases. Theme 5: Preventing bad smells from decomposed waste—a lack of municipal waste collection services results in bad smells emanating from decomposed waste. Theme 6: Converting waste into useful items—various items are recycled to produce new products. In other parts of the world, cast-offs are donated to destitute communities. Theme 7: Managing kitchen waste—in some African countries like Egypt, communities feed kitchen waste to animals, such as pigs and dogs. In some European countries, like Sweden, kitchen waste is disposed of in drains. Theme 8: Translating useful ISPs into effective SWM—molora (ash) is mostly used for pest control in gardens and farms. In the study area, it is used for various purposes, including treating unknown diseases, as a symbol for death in the house, and others. To ensure sustainable vegetable production, waste items such as ash, bones, cans, and animal dung are added to what is referred to as “Lentloane”. This practice is employed in the study areas and by international organizations such as food and agricultural organization, UKaid and USAID. Theme 9: The disappearance of the ISPs of SWM—considering that there are no municipal waste collection services in all geographical areas, ISPs of solid waste will never disappear. Theme 10: Relevance of the ISPs of SWM for the conservation and sustainable use of natural resources—the ISPs of SWM are relevant to conservation and the sustainable use of natural resources, but education on the topic at the grassroots level is flawed in schools. Theme 11: Facilitation of cheap, effective, and sustainable community environmental cleanliness—clean-up campaigns are cheap and can be participated in by everyone, young and old, men and women, from any private or public sector. They can be organised and implemented at any time during the year. Theme 12: Exhibition of cultural beliefs and values through the practice of the indigenous systems of SWM—exhibiting cultural beliefs and values by engaging in ISPs of SWM is an indication of shared collective values by a specific community. The ISPs of SWM will never disappear because, for indigenous communities, the practice is culturally accepted, people can manage it, and people do not have the support of municipal authorities for waste collection. The ISPs are relevant for conservation and the sustainable use of natural resources; however, basic education seems to be a can of worms. Therefore, the problem in the study areas is not ISPs of SWM, per se, but a lack of education and training to community members by authorities in Lesotho on how to engage positively in systems and practices of SWM. The environment can simultaneously be conserved and upheld through indigenous methods of waste management.

A critical analysis of ISPs of SWM in the rural communities of Maseru, Lesotho, and its impact on the environment and human wellbeing is significant to identify appropriate control measures for the following reasons: There is a need for authorities in Lesotho to plan for SWM in the rural areas surrounding Maseru City. To help protect human wellbeing and land from pollution and degradation, regulatory measures (specifically municipal by-laws) must be established to control the effects of inappropriate handling and disposal of solid waste in the MKRC. Current ISPs of waste disposal are inappropriate and may pose a threat to human health and the environment; appropriate waste management methods must be understood and adopted by all communities in the MKRC, and education on this topic should be provided, as the literacy rate is low in the study areas. Ultimately, appropriate SWM methods should be sustainable and benefit the MKRC communities in the long term. It is therefore important to investigate and use municipal by-laws since SWM tools may be relevant to address the issue of inappropriate systems and practices of SWM. The study will contribute information and knowledge towards rural planning in SWM and sustainable development in the MKRC. Newly developed rural municipal by-laws could be used to promote good waste management programmes that mitigate adverse effects on human wellbeing and the environment. Literature has shown that rural areas in Lesotho are not provided with municipal solid waste services; therefore, this study was the first research conducted to investigate whether municipal by-laws can be developed and implemented in the rural areas surrounding Maseru to offer alternative SWM systems and practices. This alternative could be achieved by incorporating the waste management hierarchy into ISPs. The analysis of the waste management systems in question will promote a valuable understanding of how ISPs of SWM could contribute to maintaining human wellbeing and the environment. Furthermore, to empower the MKRC communities, this study will provide insight and information on the process of ISPs so they can seek solutions for their problems. Using the results from this study, authorities in Lesotho may leverage findings to develop ISP improvement programmes by initiating cultural, values and belief factors that contribute to maintaining the traditional practices the MKRC communities prefer. Data collected from this study will be utilised to help policymakers in Lesotho make decisions on the most appropriate indigenous SWM systems and practices required according to the MKRC’s geographical location. The communities will be better equipped to accept that changes are needed to develop the rural areas in terms of SWM. They will also be encouraged to take the lead in rural policy development to ensure rural communities receive municipal waste collection services. This study may also benefit the MKRC communities by providing information on job creation opportunities, such as using stored waste to make new items; for example, plastic hats and chairs made from old tyres.

## Figures and Tables

**Figure 1 ijerph-19-11654-f001:**
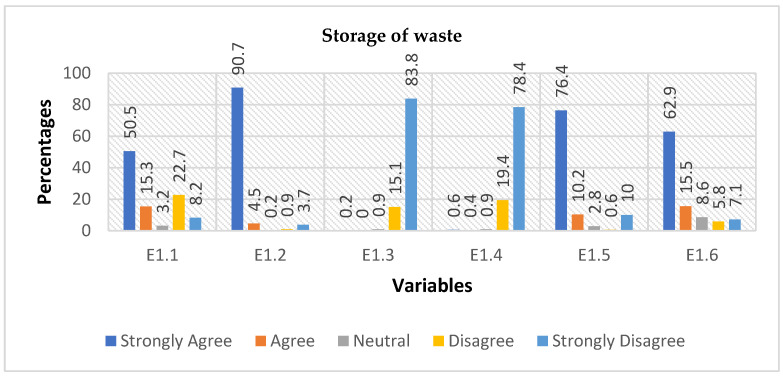
The illustration of how different types of waste can be useful to respondents if they keep it in their premises to be used in future. Variables are questions: E1.1 = Storage of waste, E1.2 = Cow dung as fuel, E1.3 = Cow dung for generator, E1.4 = Cow dung for lighting, E1.5 = Decomposed waste for manure, E1.6 = Production of new items.

**Figure 2 ijerph-19-11654-f002:**
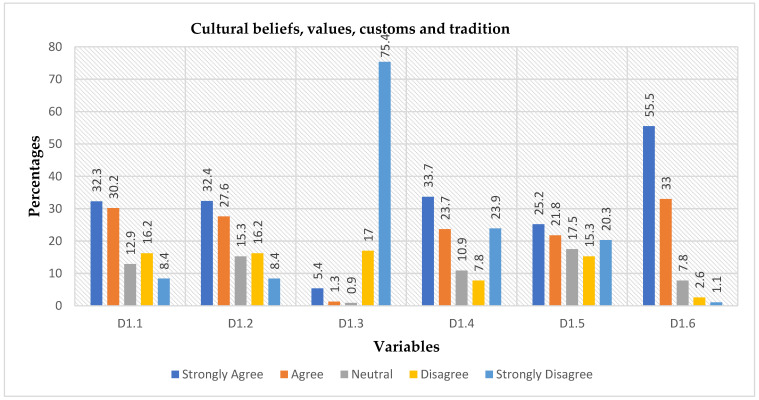
The illustration of how respondents incorporate their beliefs, values, practices, natural resources, mores, customs, rituals, and tradition to indigenous way of managing waste. Variables are questions: D1.1 = Cultural beliefs, D1.2 = Cultural values, D1.3 = Waste on the ground, D1.4 = Natural resources, D1.5 = Cultural mores, D1.6 = Customs, rituals, and tradition.

**Figure 3 ijerph-19-11654-f003:**
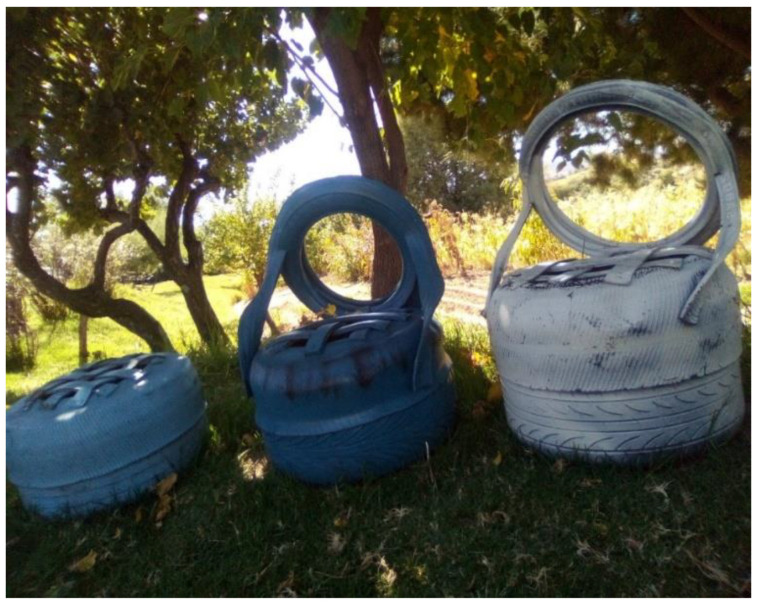
The illustration of old tyres converted into chairs in Matsieng Constituency. Picture taken by Teboho Maretlane in May 2021.

**Figure 4 ijerph-19-11654-f004:**
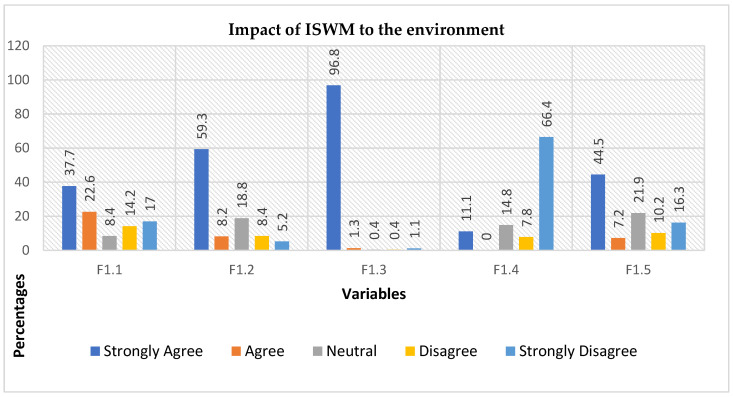
The illustration of various negative consequences that can be caused by poor management of indigenous solid waste. Variables are questions: F1.1 = Damage to the environment, F1.2 = Water and land pollution, F1.3 = Improving indigenous SW, F1.4 = Greenhouse generation, F1.5 = Leachate contamination.

**Figure 5 ijerph-19-11654-f005:**
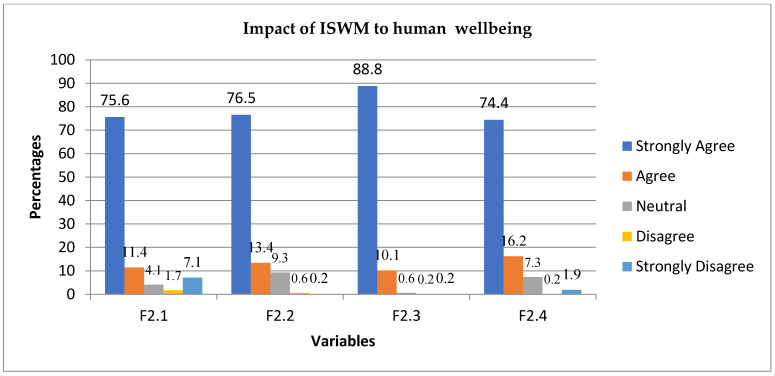
The illustration of how respondents perceive the impact of indigenous systems of waste management on human wellbeing. Variables are questions: F2.1 = Occurrence of infectious diseases, F2.2 = Rodents and mosquitoes, F2.3 = Effects on human beings, F2.4 = Ignorance.

**Figure 6 ijerph-19-11654-f006:**
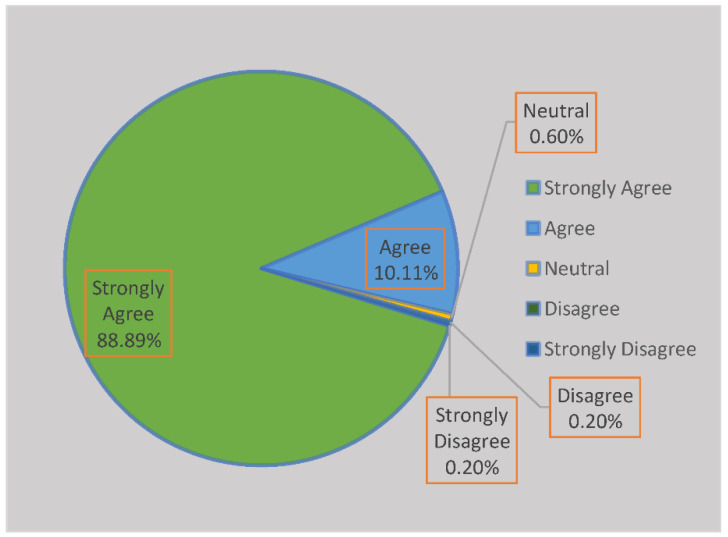
The illustration of how respondents understood the concept of solid waste management and how it can cause strong, unpleasant odour.

**Table 1 ijerph-19-11654-t001:** The illustration of Lesotho life expectancy data (2012–2021).

Year	Life Expectancy	Growth Rate
2021	54.79	0.79%
2020	54.37	0.79%
2019	53.94	0.80%
2018	53.51	2.04%
2017	52.44	2.09%
2016	51.37	2.13%
2015	50.29	2.18%
2014	49.22	2.23%
2013	48.15	2.16%
2012	47.12	2.20%

Source: [5].

**Table 2 ijerph-19-11654-t002:** Summary data on the estimated population and sample size of the communities in rural areas of Maseru that were studied.

	Rural Community	Estimated Total Population	Sample Size
Matsieng constituency	Ha Leutsoa	386	39
	Ha Moruthoane	312	31
	Kholokoe	735	74
	Ha Ramabele	212	21
	Ha Rantsilonyane	284	28
	Ha Mphafi	98	10
	Aupolasi Mahloenyeng	160	16
	Sub-total	2187	219
Koro-Koro constituency	Ha Phohleli	355	36
	Phuleng Ha Makhalanyane	208	21
	Ha Maja	569	57
	Ha Sekete	260	26
	Ha Mofoka	346	35
	Molumong Ha Mofoka	98	10
	Aupolasi Ha Mofoka	391	39
	Sub-total	2227	224
Rothe constituency	Ha Mokaoli	573	57
	Ha Thlakanelo	365	37
	Ha Rasekoai	394	39
	Mahuu	270	27
	Leralleng	353	35
	Masite Nek	548	55
	Sub-total	2503	250
Total		6917	693

Source: [5].

**Table 3 ijerph-19-11654-t003:** A summary of the demographic factors of the participants (gender, nationality, language, age group, education, marital status, employments status and length of stay).

		Frequency	Percent	Valid Percent
Gender	Male	173	37.3	38.1
	Female	281	60.6	61.9
Nationality	Mosotho	439	94.6	99.8
	Other	1	0.2	0.2
Language	Sesotho	444	95.7	98.2
	English	7	1.5	1.5
	Other	1	0.2	0.2
Age group	21–35	77	16.6	17.1
	36–45	94	20.3	20.8
	46–55	110	23.7	24.4
	56–65	110	23.7	24.4
	>65	60	12.9	13.3
Highest level of Education				
	Tertiary	16	3.4	3.5
	High school	52	11.2	11.4
	Secondary school	64	13.8	14.0
	Primary school	155	33.4	33.9
	Never went to school	170	36.6	37.2
Marital status	Single	119	25.6	26.3
	Married	266	57.3	58.7
	Divorced	11	2.4	2.4
	Separated	57	12.3	12.6
Employment Status				
	Employed	151	32.5	34.0
	Unemployed	267	57.5	60.0
	Self-employed	26	5.6	5.9
Length of stay	<1 year	8	1.7	1.8
	1–5 years	25	5.4	5.5
	6–10 years	68	14.7	14.9
	>10 years	354	76.3	77.8

Source [20].

**Table 4 ijerph-19-11654-t004:** The illustration of solid waste generated in tonnes by different income levels in Maseru in 2011.

Type of Waste	Low Income	Middle Income	High Income
Glass	5.84	7.67	2.19
Cans	5.11	4.75	0.73
Garden refuse	0.73	6.21	1.46
Kitchen leftovers	6.21	8.03	6.21
Paper	4.75	7.67	3.65
Plastic	3.65	4.38	0.11
Other	1.10	3.29	0.37
Total	27.38	41.98	14.71

Source: [5].

**Table 5 ijerph-19-11654-t005:** The illustration of types of waste observed in the MKRC by constituency.

Types of Waste Observed in the MKRC													
Constituency	Corn stalk	Corrugated Iron Sheet	Plastics	Tin	Fly Ash	Glass	Fabric	Tyres	Wood	Paper	Steel	Cow-dung	Garden waste
Matsieng	Yes	Yes	Yes	Yes	Yes	Yes	Yes	Yes	Yes	Yes	Yes	Yes	Yes
Koro-Koro	Yes	Yes	Yes	Yes	Yes	Yes	Yes	Yes	Yes	Yes	Yes	Yes	Yes
Rothe	Yes	No	Yes	Yes	Yes	Yes	Yes	No	Yes	Yes	Yes	No	No

Source: [22].

## Data Availability

Not applicable.

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
