# Peer review of "A Critical Analysis of Indigenous Systems and Practices of Solid Waste Management in Rural Communities: The Case of Maseru in Lesotho"

_ijerph, 2022, doi:10.3390/ijerph191811654_

Round 1

Reviewer 1 Report (Previous Reviewer 1)

The authors have appropriately incorporated some suggested changes. At this stage, I only have a few rather minor suggestions that should be processed before the paper can be accepted.

(i) There should be one (very last) paragraph in the Introduction describing the structure of the paper.

(ii) The readability of Figure 5 is quite low; please fix the numbers.

(iii) Labels of the figures: there should rather be „Figure 4: The illustration of…“ instead of „Figure 4 illustrates…“. Please fix this among all the figures.

(iv) Abbreviations (page 15): there should be an explanation of each of the abbreviations used.

Author Response

Response to Reviewer 1 Comments

Point 1: There should be one (very last paragraph in the introduction describing the structure of the paper.

Response 1: In this article, we present the methodology employed in this study and pay special attention to explanation of the methodological approach, description of data collection method. We explain quantitative descriptive design, ethnographic design, and direct observation design. We further discuss data collection and data analysis. Next, we present the results of the study and provide the discussion where we focus on community perceived understanding of indigenous practices of SWM in the MKRC, community perceptions of the impact of indigenous solid waste on the environment, community perceptions of the impact of indigenous solid waste on human wellbeing. This is followed by types of solid waste generated in the MKRC. Next is gaps in knowledge and then conclusion.

Point 2: The readability of figure 5 is quite low; please fix the numbers.

Response 2: We have fixed the numbers and they are now visible.

Point 3: Labels of the figure: there should rather be “Figure 4: The illustration of…..’instead of “Figure 4 illustrates….” Please fix this among all the figures.

Response 3: We have rephrased “illustrates” to be “ The illustration of ….” among all figures.

Point 4: Abbreviations (page 15): there should be an explanation of each of the abbreviations used.

Response 4: All abbreviations have been explained as follows:

Ag                                                        Silver

As                                                         Arsenic

Ba                                                         Barium

Cd                                                        Cadmium

COVID-19                                            Corona Virus Disease 19

Cu                                                        Copper

Fe                                                        Iron

GHG                                                    Greenhouse gas

Hg                                                        Mercury

ISPs                                                     Indigenous systems and practices

ISWM.                                                 Indigenous solid waste management

MKRC                                                 Matsieng, Koro-Koro and Rothe constituencies

Mn                                                       Manganese

Ni                                                        Nickel

Pb                                                        Lead

PHC                                                     Petroleum hydrocarbons

SWM                                                   Solid waste management

Zn                                                        Zinc

Response to Reviewer 1 Comments

Point 1: There should be one (very last paragraph in the introduction describing the structure of the paper.

Response 1: In this article, we present the methodology employed in this study and pay special attention to explanation of the methodological approach, description of data collection method. We explain quantitative descriptive design, ethnographic design, and direct observation design. We further discuss data collection and data analysis. Next, we present the results of the study and provide the discussion where we focus on community perceived understanding of indigenous practices of SWM in the MKRC, community perceptions of the impact of indigenous solid waste on the environment, community perceptions of the impact of indigenous solid waste on human wellbeing. This is followed by types of solid waste generated in the MKRC. Next is gaps in knowledge and then conclusion.

Point 2: The readability of figure 5 is quite low; please fix the numbers.

Response 2: We have fixed the numbers and they are now visible.

Point 3: Labels of the figure: there should rather be “Figure 4: The illustration of…..’instead of “Figure 4 illustrates….” Please fix this among all the figures.

Response 3: We have rephrased “illustrates” to be “ The illustration of ….” among all figures.

Point 4: Abbreviations (page 15): there should be an explanation of each of the abbreviations used.

Response 4: All abbreviations have been explained as follows:

Ag                                                        Silver

As                                                         Arsenic

Ba                                                         Barium

Cd                                                        Cadmium

COVID-19                                            Corona Virus Disease 19

Cu                                                        Copper

Fe                                                        Iron

GHG                                                    Greenhouse gas

Hg                                                        Mercury

ISPs                                                     Indigenous systems and practices

ISWM.                                                 Indigenous solid waste management

MKRC                                                 Matsieng, Koro-Koro and Rothe constituencies

Mn                                                       Manganese

Ni                                                        Nickel

Pb                                                        Lead

PHC                                                     Petroleum hydrocarbons

SWM                                                   Solid waste management

Zn                                                        Zinc

Response to Reviewer 1 Comments

Point 1: There should be one (very last paragraph in the introduction describing the structure of the paper.

Response 1: In this article, we present the methodology employed in this study and pay special attention to explanation of the methodological approach, description of data collection method. We explain quantitative descriptive design, ethnographic design, and direct observation design. We further discuss data collection and data analysis. Next, we present the results of the study and provide the discussion where we focus on community perceived understanding of indigenous practices of SWM in the MKRC, community perceptions of the impact of indigenous solid waste on the environment, community perceptions of the impact of indigenous solid waste on human wellbeing. This is followed by types of solid waste generated in the MKRC. Next is gaps in knowledge and then conclusion.

Point 2: The readability of figure 5 is quite low; please fix the numbers.

Response 2: We have fixed the numbers and they are now visible.

Point 3: Labels of the figure: there should rather be “Figure 4: The illustration of…..’instead of “Figure 4 illustrates….” Please fix this among all the figures.

Response 3: We have rephrased “illustrates” to be “ The illustration of ….” among all figures.

Point 4: Abbreviations (page 15): there should be an explanation of each of the abbreviations used.

Response 4: All abbreviations have been explained as follows:

Ag                                                        Silver

As                                                         Arsenic

Ba                                                         Barium

Cd                                                        Cadmium

COVID-19                                            Corona Virus Disease 19

Cu                                                        Copper

Fe                                                        Iron

GHG                                                    Greenhouse gas

Hg                                                        Mercury

ISPs                                                     Indigenous systems and practices

ISWM.                                                 Indigenous solid waste management

MKRC                                                 Matsieng, Koro-Koro and Rothe constituencies

Mn                                                       Manganese

Ni                                                        Nickel

Pb                                                        Lead

PHC                                                     Petroleum hydrocarbons

SWM                                                   Solid waste management

Zn                                                        Zinc

Response to Reviewer 1 Comments

Point 1: There should be one (very last paragraph in the introduction describing the structure of the paper.

Response 1: In this article, we present the methodology employed in this study and pay special attention to explanation of the methodological approach, description of data collection method. We explain quantitative descriptive design, ethnographic design, and direct observation design. We further discuss data collection and data analysis. Next, we present the results of the study and provide the discussion where we focus on community perceived understanding of indigenous practices of SWM in the MKRC, community perceptions of the impact of indigenous solid waste on the environment, community perceptions of the impact of indigenous solid waste on human wellbeing. This is followed by types of solid waste generated in the MKRC. Next is gaps in knowledge and then conclusion.

Point 2: The readability of figure 5 is quite low; please fix the numbers.

Response 2: We have fixed the numbers and they are now visible.

Point 3: Labels of the figure: there should rather be “Figure 4: The illustration of…..’instead of “Figure 4 illustrates….” Please fix this among all the figures.

Response 3: We have rephrased “illustrates” to be “ The illustration of ….” among all figures.

Point 4: Abbreviations (page 15): there should be an explanation of each of the abbreviations used.

Response 4: All abbreviations have been explained as follows:

Ag                                                        Silver

As                                                         Arsenic

Ba                                                         Barium

Cd                                                        Cadmium

COVID-19                                            Corona Virus Disease 19

Cu                                                        Copper

Fe                                                        Iron

GHG                                                    Greenhouse gas

Hg                                                        Mercury

ISPs                                                     Indigenous systems and practices

ISWM.                                                 Indigenous solid waste management

MKRC                                                 Matsieng, Koro-Koro and Rothe constituencies

Mn                                                       Manganese

Ni                                                        Nickel

Pb                                                        Lead

PHC                                                     Petroleum hydrocarbons

SWM                                                   Solid waste management

Zn                                                        Zinc

Response to Reviewer 1 Comments

Point 1: There should be one (very last paragraph in the introduction describing the structure of the paper.

Response 1: In this article, we present the methodology employed in this study and pay special attention to explanation of the methodological approach, description of data collection method. We explain quantitative descriptive design, ethnographic design, and direct observation design. We further discuss data collection and data analysis. Next, we present the results of the study and provide the discussion where we focus on community perceived understanding of indigenous practices of SWM in the MKRC, community perceptions of the impact of indigenous solid waste on the environment, community perceptions of the impact of indigenous solid waste on human wellbeing. This is followed by types of solid waste generated in the MKRC. Next is gaps in knowledge and then conclusion.

Point 2: The readability of figure 5 is quite low; please fix the numbers.

Response 2: We have fixed the numbers and they are now visible.

Point 3: Labels of the figure: there should rather be “Figure 4: The illustration of…..’instead of “Figure 4 illustrates….” Please fix this among all the figures.

Response 3: We have rephrased “illustrates” to be “ The illustration of ….” among all figures.

Point 4: Abbreviations (page 15): there should be an explanation of each of the abbreviations used.

Response 4: All abbreviations have been explained as follows:

Ag                                                        Silver

As                                                         Arsenic

Ba                                                         Barium

Cd                                                        Cadmium

COVID-19                                            Corona Virus Disease 19

Cu                                                        Copper

Fe                                                        Iron

GHG                                                    Greenhouse gas

Hg                                                        Mercury

ISPs                                                     Indigenous systems and practices

ISWM.                                                 Indigenous solid waste management

MKRC                                                 Matsieng, Koro-Koro and Rothe constituencies

Mn                                                       Manganese

Ni                                                        Nickel

Pb                                                        Lead

PHC                                                     Petroleum hydrocarbons

SWM                                                   Solid waste management

Zn                                                        Zinc

Response to Reviewer 1 Comments

Point 1: There should be one (very last paragraph in the introduction describing the structure of the paper.

Response 1: In this article, we present the methodology employed in this study and pay special attention to explanation of the methodological approach, description of data collection method. We explain quantitative descriptive design, ethnographic design, and direct observation design. We further discuss data collection and data analysis. Next, we present the results of the study and provide the discussion where we focus on community perceived understanding of indigenous practices of SWM in the MKRC, community perceptions of the impact of indigenous solid waste on the environment, community perceptions of the impact of indigenous solid waste on human wellbeing. This is followed by types of solid waste generated in the MKRC. Next is gaps in knowledge and then conclusion.

Point 2: The readability of figure 5 is quite low; please fix the numbers.

Response 2: We have fixed the numbers and they are now visible.

Point 3: Labels of the figure: there should rather be “Figure 4: The illustration of…..’instead of “Figure 4 illustrates….” Please fix this among all the figures.

Response 3: We have rephrased “illustrates” to be “ The illustration of ….” among all figures.

Point 4: Abbreviations (page 15): there should be an explanation of each of the abbreviations used.

Response 4: All abbreviations have been explained as follows:

Ag                                                        Silver

As                                                         Arsenic

Ba                                                         Barium

Cd                                                        Cadmium

COVID-19                                            Corona Virus Disease 19

Cu                                                        Copper

Fe                                                        Iron

GHG                                                    Greenhouse gas

Hg                                                        Mercury

ISPs                                                     Indigenous systems and practices

ISWM.                                                 Indigenous solid waste management

MKRC                                                 Matsieng, Koro-Koro and Rothe constituencies

Mn                                                       Manganese

Ni                                                        Nickel

Pb                                                        Lead

PHC                                                     Petroleum hydrocarbons

SWM                                                   Solid waste management

Zn                                                        Zinc

Response to Reviewer 1 Comments

Point 1: There should be one (very last paragraph in the introduction describing the structure of the paper.

Response 1: In this article, we present the methodology employed in this study and pay special attention to explanation of the methodological approach, description of data collection method. We explain quantitative descriptive design, ethnographic design, and direct observation design. We further discuss data collection and data analysis. Next, we present the results of the study and provide the discussion where we focus on community perceived understanding of indigenous practices of SWM in the MKRC, community perceptions of the impact of indigenous solid waste on the environment, community perceptions of the impact of indigenous solid waste on human wellbeing. This is followed by types of solid waste generated in the MKRC. Next is gaps in knowledge and then conclusion.

Point 2: The readability of figure 5 is quite low; please fix the numbers.

Response 2: We have fixed the numbers and they are now visible.

Point 3: Labels of the figure: there should rather be “Figure 4: The illustration of…..’instead of “Figure 4 illustrates….” Please fix this among all the figures.

Response 3: We have rephrased “illustrates” to be “ The illustration of ….” among all figures.

Point 4: Abbreviations (page 15): there should be an explanation of each of the abbreviations used.

Response 4: All abbreviations have been explained as follows:

Ag                                                        Silver

As                                                         Arsenic

Ba                                                         Barium

Cd                                                        Cadmium

COVID-19                                            Corona Virus Disease 19

Cu                                                        Copper

Fe                                                        Iron

GHG                                                    Greenhouse gas

Hg                                                        Mercury

ISPs                                                     Indigenous systems and practices

ISWM.                                                 Indigenous solid waste management

MKRC                                                 Matsieng, Koro-Koro and Rothe constituencies

Mn                                                       Manganese

Ni                                                        Nickel

Pb                                                        Lead

PHC                                                     Petroleum hydrocarbons

SWM                                                   Solid waste management

Zn                                                        Zinc

Response to Reviewer 1 Comments

Point 1: There should be one (very last paragraph in the introduction describing the structure of the paper.

Response 1: In this article, we present the methodology employed in this study and pay special attention to explanation of the methodological approach, description of data collection method. We explain quantitative descriptive design, ethnographic design, and direct observation design. We further discuss data collection and data analysis. Next, we present the results of the study and provide the discussion where we focus on community perceived understanding of indigenous practices of SWM in the MKRC, community perceptions of the impact of indigenous solid waste on the environment, community perceptions of the impact of indigenous solid waste on human wellbeing. This is followed by types of solid waste generated in the MKRC. Next is gaps in knowledge and then conclusion.

Point 2: The readability of figure 5 is quite low; please fix the numbers.

Response 2: We have fixed the numbers and they are now visible.

Point 3: Labels of the figure: there should rather be “Figure 4: The illustration of…..’instead of “Figure 4 illustrates….” Please fix this among all the figures.

Response 3: We have rephrased “illustrates” to be “ The illustration of ….” among all figures.

Point 4: Abbreviations (page 15): there should be an explanation of each of the abbreviations used.

Response 4: All abbreviations have been explained as follows:

Ag                                                        Silver

As                                                         Arsenic

Ba                                                         Barium

Cd                                                        Cadmium

COVID-19                                            Corona Virus Disease 19

Cu                                                        Copper

Fe                                                        Iron

GHG                                                    Greenhouse gas

Hg                                                        Mercury

ISPs                                                     Indigenous systems and practices

ISWM.                                                 Indigenous solid waste management

MKRC                                                 Matsieng, Koro-Koro and Rothe constituencies

Mn                                                       Manganese

Ni                                                        Nickel

Pb                                                        Lead

PHC                                                     Petroleum hydrocarbons

SWM                                                   Solid waste management

Zn                                                        Zinc

Response to Reviewer 1 Comments

Point 1: There should be one (very last paragraph in the introduction describing the structure of the paper.

Response 1: In this article, we present the methodology employed in this study and pay special attention to explanation of the methodological approach, description of data collection method. We explain quantitative descriptive design, ethnographic design, and direct observation design. We further discuss data collection and data analysis. Next, we present the results of the study and provide the discussion where we focus on community perceived understanding of indigenous practices of SWM in the MKRC, community perceptions of the impact of indigenous solid waste on the environment, community perceptions of the impact of indigenous solid waste on human wellbeing. This is followed by types of solid waste generated in the MKRC. Next is gaps in knowledge and then conclusion.

Point 2: The readability of figure 5 is quite low; please fix the numbers.

Response 2: We have fixed the numbers and they are now visible.

Point 3: Labels of the figure: there should rather be “Figure 4: The illustration of…..’instead of “Figure 4 illustrates….” Please fix this among all the figures.

Response 3: We have rephrased “illustrates” to be “ The illustration of ….” among all figures.

Point 4: Abbreviations (page 15): there should be an explanation of each of the abbreviations used.

Response 4: All abbreviations have been explained as follows:

Ag                                                        Silver

As                                                         Arsenic

Ba                                                         Barium

Cd                                                        Cadmium

COVID-19                                            Corona Virus Disease 19

Cu                                                        Copper

Fe                                                        Iron

GHG                                                    Greenhouse gas

Hg                                                        Mercury

ISPs                                                     Indigenous systems and practices

ISWM.                                                 Indigenous solid waste management

MKRC                                                 Matsieng, Koro-Koro and Rothe constituencies

Mn                                                       Manganese

Ni                                                        Nickel

Pb                                                        Lead

PHC                                                     Petroleum hydrocarbons

SWM                                                   Solid waste management

Zn                                                        Zinc

Response to Reviewer 1 Comments

Point 1: There should be one (very last paragraph in the introduction describing the structure of the paper.

Response 1: In this article, we present the methodology employed in this study and pay special attention to explanation of the methodological approach, description of data collection method. We explain quantitative descriptive design, ethnographic design, and direct observation design. We further discuss data collection and data analysis. Next, we present the results of the study and provide the discussion where we focus on community perceived understanding of indigenous practices of SWM in the MKRC, community perceptions of the impact of indigenous solid waste on the environment, community perceptions of the impact of indigenous solid waste on human wellbeing. This is followed by types of solid waste generated in the MKRC. Next is gaps in knowledge and then conclusion.

Point 2: The readability of figure 5 is quite low; please fix the numbers.

Response 2: We have fixed the numbers and they are now visible.

Point 3: Labels of the figure: there should rather be “Figure 4: The illustration of…..’instead of “Figure 4 illustrates….” Please fix this among all the figures.

Response 3: We have rephrased “illustrates” to be “ The illustration of ….” among all figures.

Point 4: Abbreviations (page 15): there should be an explanation of each of the abbreviations used.

Response 4: All abbreviations have been explained as follows:

Ag                                                        Silver

As                                                         Arsenic

Ba                                                         Barium

Cd                                                        Cadmium

COVID-19                                            Corona Virus Disease 19

Cu                                                        Copper

Fe                                                        Iron

GHG                                                    Greenhouse gas

Hg                                                        Mercury

ISPs                                                     Indigenous systems and practices

ISWM.                                                 Indigenous solid waste management

MKRC                                                 Matsieng, Koro-Koro and Rothe constituencies

Mn                                                       Manganese

Ni                                                        Nickel

Pb                                                        Lead

PHC                                                     Petroleum hydrocarbons

SWM                                                   Solid waste management

Zn                                                        Zinc

Response to Reviewer 1 Comments

Point 1: There should be one (very last paragraph in the introduction describing the structure of the paper.

Response 1: In this article, we present the methodology employed in this study and pay special attention to explanation of the methodological approach, description of data collection method. We explain quantitative descriptive design, ethnographic design, and direct observation design. We further discuss data collection and data analysis. Next, we present the results of the study and provide the discussion where we focus on community perceived understanding of indigenous practices of SWM in the MKRC, community perceptions of the impact of indigenous solid waste on the environment, community perceptions of the impact of indigenous solid waste on human wellbeing. This is followed by types of solid waste generated in the MKRC. Next is gaps in knowledge and then conclusion.

Point 2: The readability of figure 5 is quite low; please fix the numbers.

Response 2: We have fixed the numbers and they are now visible.

Point 3: Labels of the figure: there should rather be “Figure 4: The illustration of…..’instead of “Figure 4 illustrates….” Please fix this among all the figures.

Response 3: We have rephrased “illustrates” to be “ The illustration of ….” among all figures.

Point 4: Abbreviations (page 15): there should be an explanation of each of the abbreviations used.

Response 4: All abbreviations have been explained as follows:

Ag                                                        Silver

As                                                         Arsenic

Ba                                                         Barium

Cd                                                        Cadmium

COVID-19                                            Corona Virus Disease 19

Cu                                                        Copper

Fe                                                        Iron

GHG                                                    Greenhouse gas

Hg                                                        Mercury

ISPs                                                     Indigenous systems and practices

ISWM.                                                 Indigenous solid waste management

MKRC                                                 Matsieng, Koro-Koro and Rothe constituencies

Mn                                                       Manganese

Ni                                                        Nickel

Pb                                                        Lead

PHC                                                     Petroleum hydrocarbons

SWM                                                   Solid waste management

Zn                                                        Zinc

Response to Reviewer 1 Comments

Point 1: There should be one (very last paragraph in the introduction describing the structure of the paper.

Response 1: In this article, we present the methodology employed in this study and pay special attention to explanation of the methodological approach, description of data collection method. We explain quantitative descriptive design, ethnographic design, and direct observation design. We further discuss data collection and data analysis. Next, we present the results of the study and provide the discussion where we focus on community perceived understanding of indigenous practices of SWM in the MKRC, community perceptions of the impact of indigenous solid waste on the environment, community perceptions of the impact of indigenous solid waste on human wellbeing. This is followed by types of solid waste generated in the MKRC. Next is gaps in knowledge and then conclusion.

Point 2: The readability of figure 5 is quite low; please fix the numbers.

Response 2: We have fixed the numbers and they are now visible.

Point 3: Labels of the figure: there should rather be “Figure 4: The illustration of…..’instead of “Figure 4 illustrates….” Please fix this among all the figures.

Response 3: We have rephrased “illustrates” to be “ The illustration of ….” among all figures.

Point 4: Abbreviations (page 15): there should be an explanation of each of the abbreviations used.

Response 4: All abbreviations have been explained as follows:

Ag                                                        Silver

As                                                         Arsenic

Ba                                                         Barium

Cd                                                        Cadmium

COVID-19                                            Corona Virus Disease 19

Cu                                                        Copper

Fe                                                        Iron

GHG                                                    Greenhouse gas

Hg                                                        Mercury

ISPs                                                     Indigenous systems and practices

ISWM.                                                 Indigenous solid waste management

MKRC                                                 Matsieng, Koro-Koro and Rothe constituencies

Mn                                                       Manganese

Ni                                                        Nickel

Pb                                                        Lead

PHC                                                     Petroleum hydrocarbons

SWM                                                   Solid waste management

Zn                                                        Zinc

Response to Reviewer 1 Comments

Point 1: There should be one (very last paragraph in the introduction describing the structure of the paper.

Response 1: In this article, we present the methodology employed in this study and pay special attention to explanation of the methodological approach, description of data collection method. We explain quantitative descriptive design, ethnographic design, and direct observation design. We further discuss data collection and data analysis. Next, we present the results of the study and provide the discussion where we focus on community perceived understanding of indigenous practices of SWM in the MKRC, community perceptions of the impact of indigenous solid waste on the environment, community perceptions of the impact of indigenous solid waste on human wellbeing. This is followed by types of solid waste generated in the MKRC. Next is gaps in knowledge and then conclusion.

Point 2: The readability of figure 5 is quite low; please fix the numbers.

Response 2: We have fixed the numbers and they are now visible.

Point 3: Labels of the figure: there should rather be “Figure 4: The illustration of…..’instead of “Figure 4 illustrates….” Please fix this among all the figures.

Response 3: We have rephrased “illustrates” to be “ The illustration of ….” among all figures.

Point 4: Abbreviations (page 15): there should be an explanation of each of the abbreviations used.

Response 4: All abbreviations have been explained as follows:

Ag                                                        Silver

As                                                         Arsenic

Ba                                                         Barium

Cd                                                        Cadmium

COVID-19                                            Corona Virus Disease 19

Cu                                                        Copper

Fe                                                        Iron

GHG                                                    Greenhouse gas

Hg                                                        Mercury

ISPs                                                     Indigenous systems and practices

ISWM.                                                 Indigenous solid waste management

MKRC                                                 Matsieng, Koro-Koro and Rothe constituencies

Mn                                                       Manganese

Ni                                                        Nickel

Pb                                                        Lead

PHC                                                     Petroleum hydrocarbons

SWM                                                   Solid waste management

Zn                                                        Zinc

Response to Reviewer 1 Comments

Point 1: There should be one (very last paragraph in the introduction describing the structure of the paper.

Response 1: In this article, we present the methodology employed in this study and pay special attention to explanation of the methodological approach, description of data collection method. We explain quantitative descriptive design, ethnographic design, and direct observation design. We further discuss data collection and data analysis. Next, we present the results of the study and provide the discussion where we focus on community perceived understanding of indigenous practices of SWM in the MKRC, community perceptions of the impact of indigenous solid waste on the environment, community perceptions of the impact of indigenous solid waste on human wellbeing. This is followed by types of solid waste generated in the MKRC. Next is gaps in knowledge and then conclusion.

Point 2: The readability of figure 5 is quite low; please fix the numbers.

Response 2: We have fixed the numbers and they are now visible.

Point 3: Labels of the figure: there should rather be “Figure 4: The illustration of…..’instead of “Figure 4 illustrates….” Please fix this among all the figures.

Response 3: We have rephrased “illustrates” to be “ The illustration of ….” among all figures.

Point 4: Abbreviations (page 15): there should be an explanation of each of the abbreviations used.

Response 4: All abbreviations have been explained as follows:

Ag                                                        Silver

As                                                         Arsenic

Ba                                                         Barium

Cd                                                        Cadmium

COVID-19                                            Corona Virus Disease 19

Cu                                                        Copper

Fe                                                        Iron

GHG                                                    Greenhouse gas

Hg                                                        Mercury

ISPs                                                     Indigenous systems and practices

ISWM.                                                 Indigenous solid waste management

MKRC                                                 Matsieng, Koro-Koro and Rothe constituencies

Mn                                                       Manganese

Ni                                                        Nickel

Pb                                                        Lead

PHC                                                     Petroleum hydrocarbons

SWM                                                   Solid waste management

Zn                                                        Zinc

Response to Reviewer 1 Comments

Point 1: There should be one (very last paragraph in the introduction describing the structure of the paper.

Response 1: In this article, we present the methodology employed in this study and pay special attention to explanation of the methodological approach, description of data collection method. We explain quantitative descriptive design, ethnographic design, and direct observation design. We further discuss data collection and data analysis. Next, we present the results of the study and provide the discussion where we focus on community perceived understanding of indigenous practices of SWM in the MKRC, community perceptions of the impact of indigenous solid waste on the environment, community perceptions of the impact of indigenous solid waste on human wellbeing. This is followed by types of solid waste generated in the MKRC. Next is gaps in knowledge and then conclusion.

Point 2: The readability of figure 5 is quite low; please fix the numbers.

Response 2: We have fixed the numbers and they are now visible.

Point 3: Labels of the figure: there should rather be “Figure 4: The illustration of…..’instead of “Figure 4 illustrates….” Please fix this among all the figures.

Response 3: We have rephrased “illustrates” to be “ The illustration of ….” among all figures.

Point 4: Abbreviations (page 15): there should be an explanation of each of the abbreviations used.

Response 4: All abbreviations have been explained as follows:

Ag                                                        Silver

As                                                         Arsenic

Ba                                                         Barium

Cd                                                        Cadmium

COVID-19                                            Corona Virus Disease 19

Cu                                                        Copper

Fe                                                        Iron

GHG                                                    Greenhouse gas

Hg                                                        Mercury

ISPs                                                     Indigenous systems and practices

ISWM.                                                 Indigenous solid waste management

MKRC                                                 Matsieng, Koro-Koro and Rothe constituencies

Mn                                                       Manganese

Ni                                                        Nickel

Pb                                                        Lead

PHC                                                     Petroleum hydrocarbons

SWM                                                   Solid waste management

Zn                                                        Zinc

Response to Reviewer 1 Comments

Point 1: There should be one (very last paragraph in the introduction describing the structure of the paper.

Response 1: In this article, we present the methodology employed in this study and pay special attention to explanation of the methodological approach, description of data collection method. We explain quantitative descriptive design, ethnographic design, and direct observation design. We further discuss data collection and data analysis. Next, we present the results of the study and provide the discussion where we focus on community perceived understanding of indigenous practices of SWM in the MKRC, community perceptions of the impact of indigenous solid waste on the environment, community perceptions of the impact of indigenous solid waste on human wellbeing. This is followed by types of solid waste generated in the MKRC. Next is gaps in knowledge and then conclusion.

Point 2: The readability of figure 5 is quite low; please fix the numbers.

Response 2: We have fixed the numbers and they are now visible.

Point 3: Labels of the figure: there should rather be “Figure 4: The illustration of…..’instead of “Figure 4 illustrates….” Please fix this among all the figures.

Response 3: We have rephrased “illustrates” to be “ The illustration of ….” among all figures.

Point 4: Abbreviations (page 15): there should be an explanation of each of the abbreviations used.

Response 4: All abbreviations have been explained as follows:

Ag                                                        Silver

As                                                         Arsenic

Ba                                                         Barium

Cd                                                        Cadmium

COVID-19                                            Corona Virus Disease 19

Cu                                                        Copper

Fe                                                        Iron

GHG                                                    Greenhouse gas

Hg                                                        Mercury

ISPs                                                     Indigenous systems and practices

ISWM.                                                 Indigenous solid waste management

MKRC                                                 Matsieng, Koro-Koro and Rothe constituencies

Mn                                                       Manganese

Ni                                                        Nickel

Pb                                                        Lead

PHC                                                     Petroleum hydrocarbons

SWM                                                   Solid waste management

Zn                                                        Zinc

Response to Reviewer 1 Comments

Point 1: There should be one (very last paragraph in the introduction describing the structure of the paper.

Response 1: In this article, we present the methodology employed in this study and pay special attention to explanation of the methodological approach, description of data collection method. We explain quantitative descriptive design, ethnographic design, and direct observation design. We further discuss data collection and data analysis. Next, we present the results of the study and provide the discussion where we focus on community perceived understanding of indigenous practices of SWM in the MKRC, community perceptions of the impact of indigenous solid waste on the environment, community perceptions of the impact of indigenous solid waste on human wellbeing. This is followed by types of solid waste generated in the MKRC. Next is gaps in knowledge and then conclusion.

Point 2: The readability of figure 5 is quite low; please fix the numbers.

Response 2: We have fixed the numbers and they are now visible.

Point 3: Labels of the figure: there should rather be “Figure 4: The illustration of…..’instead of “Figure 4 illustrates….” Please fix this among all the figures.

Response 3: We have rephrased “illustrates” to be “ The illustration of ….” among all figures.

Point 4: Abbreviations (page 15): there should be an explanation of each of the abbreviations used.

Response 4: All abbreviations have been explained as follows:

Ag                                                        Silver

As                                                         Arsenic

Ba                                                         Barium

Cd                                                        Cadmium

COVID-19                                            Corona Virus Disease 19

Cu                                                        Copper

Fe                                                        Iron

GHG                                                    Greenhouse gas

Hg                                                        Mercury

ISPs                                                     Indigenous systems and practices

ISWM.                                                 Indigenous solid waste management

MKRC                                                 Matsieng, Koro-Koro and Rothe constituencies

Mn                                                       Manganese

Ni                                                        Nickel

Pb                                                        Lead

PHC                                                     Petroleum hydrocarbons

SWM                                                   Solid waste management

Zn                                                        Zinc

Reviewer 2 Report (Previous Reviewer 3)

All my comments have been included in the new version of the manuscript. The manuscript has been significantly improved and now warrants publication in IJERPH.

Author Response

Response to Reviewer 2 Comments
Reviewer is satisfied with the new version of the manuscript and warrants publication in IJERPH

Reviewer 3 Report (New Reviewer)

Author Response

Thank you very much for your comments. Please check the attachment to find our response.

Round 2

Reviewer 3 Report (New Reviewer)

The reviewer acknowledges the efforts made by the authors to improve the quality of the manuscript. The manuscript as a whole is now more logically described and easy to follow. 

This manuscript is a resubmission of an earlier submission. The following is a list of the peer review reports and author responses from that submission.

Round 1

Reviewer 1 Report

The paper follows the authors' previous research and tries to state some quantitative results and insights into the waste management practices in the selected area in Lesotho. However, the paper lacks some scientific form and attributes.

  • Section 1: The paper has only a very short introduction section, even without any state-of-the-art information, literature review, or description of the aims and structure of the paper. Why the authors even do not cite this own paper that they follow?
  • Section 2: The authors mention "research questions "; however, I have not found any clearly stated research question that this research attempts to answer.
  • Section 3: (a) I don't understand how the questions were asked and answered. See, for example, the following: "The first question asked how they store their generated waste; 143 there were 463 responses, and out of these, 50.5% strongly agreed, 15.3% agreed, 3.2% 144 were neutral, 22.7% disagreed, while 8.2% strongly disagreed." From this, I have no idea what, e.g., "strongly agree" means. (b) The size of the figures is not unified.
  • Section 4 and 5: These sections are too short and do not involve any scientific results and information.

Reviewer 2 Report

Line 16: Need to add an “and” right before “to do”.

Line 18: There is only one objective. Therefore, “objective was” should be used instead of “objectives were”.

Line 20: What do you mean “pilot study”? What is its relationship with the current study?

Line 20: Where / what is Ha Teko? It’d better to give a brief description (in one short phrase) about it.

Line 22: “693 questionnaires” means 693 different sets of questionnaire? Or you have received 693 completed questionnaires of which the same set questions are used? Or you have sent out 693 questionnaires of which the same set questions are used?

Line 23-30: These contents should be put in the methodology instead of in the abstract. Apart from the key study aims, the abstract should contain at least the key methods (e.g. sampling method, sampling size and date, sampling locations), key results, key discussion and implications, and key conclusions. These essential contents are missing in this abstract.

Line 31: I am not sure “human wellbeing” is related to this study. The abstract does not show the linkage.

Line 37-38: I don’t understand the sentence “ISPs also supports sustainable development by researchers”. The logics is not clear here.

Line 45-46: There are obvious English errors in “……global warming….transmit diseases.” Pls re-write.

Line 34-51: The central idea is not well supported by the examples; the given examples do not articulate with the central idea. No relevant previous literatures on the topic are mentioned. The research gap is unknown.

Line 52: “Methods” should be replaced by “Methodology”.

Line 54: “findings” should be replaced by “methods”. Need to delete “(expressed in numbers)” and (expressed in words)”.

Line 55: Delete “data collection methods”.

Line 55: Primary and secondary data of what kind of information? Delete “We used primary and secondary data”.

Line 57: What kind of information did you obtain from the government documents and from the internet? What kind of quantitative descriptive data did you collect? How did you operate your observations?

Line 62-65: What is the point to put these contents here?

Line 78-79: Why families were thus selected? There is no explanation why families were selected. So, only families were selected? How about those did not have a family? Do you mean “household”?

Line 81: Why qualitative ethnographic method would help to gain acceptance by the community?

Line 106: What is the research gap? What is the research problem? These are not well introduced or explained in the previous section(s).

Line 118-119: Why it was necessary for the researchers to prepare the questions?

Line 120: What types of interview questions were you referring to?

Line 124: No need to mention “a hectic time” but you may explain how you collected data under the COVID-19 regulations. “completed” should be changed into “complete”.

Line 126: What do you mean “(n=12)” here? Does it mean that your sample size is 12 only?

Line 52-129: When, how and where did you collect the data? Did you obtain ethical approval from your research unit? If yes, it is on what methods, when did you obtain the approval, and what research unit approved it? These information are essential but are missing here.

Line 113 – 129: For the quantitative data collection, who was your sampling target? What was the target sample size? Regarding the qualitative data collection, what selection criteria on your target population? What was the sample size and how did you determine the sample size? You may consider to combine this section with the other sections under “Methods”.

Line 131: Outliers may happen because of insufficient collected data. How did you determine outliers? How did you treat the missing data?

Line 133 and 134: What do you mean “Data were analysed in terms of ISPs of SWM”? For quantitative analysis, did you use any software(s) to enter the data and run statistical analysis? Did you operate any statistical analysis?

Line 134-135: What do you mean “percentage was used to communicate how a group of participants related to a larger group of participants”?

Line 136: Delete “to conduct a qualitative analysis”.

Line 162: No texts are referring to Figure 1. The axis titles and units are missing. Need to delete the title on the graph as the graph already has a title at the bottom. The figure title does not give enough details – for example, storage of waste of where / who? Is it only about one kind of waste?

Line 171-201: Why did you say so? What is the evidence? What are the findings that support these views? There is no/weak linkage between your results and discussion. You need to specify which finding leads to what implication(s).

Line 202: Same as other figures, no texts are referring to this figure. The axis titles and units are missing. Need to delete the title on the graph as the graph already has a title at the bottom.

Line 212-220: Why did you say so? What is the evidence? What are the findings that support these views? There is no/weak linkage between your results and discussion. You need to specify which finding leads to what implication(s).

Line 222: Same as other figures, no texts are referring to this figure.

Line 225: Who were “they”?

Line 241: Same as other figures, no texts are referring to this figure. The axis titles and units are missing. Need to delete the title on the graph as the graph already has a title at the bottom.

Line 249-294: Why did you say so? What is the evidence? What are the findings that support these views? There is no/weak linkage between your results and discussion. You need to specify which finding leads to what implication(s).

Line 308: Same as other figures, no texts are referring to this figure. The axis titles and units are missing. Need to delete the title on the graph as the graph already has a title at the bottom.

Line 317: Same as other figures, no texts are referring to this figure. Need to delete the title on the graph. What is “c2.4”? The figure title does not clearly indicate what the graph is presenting.

Line 318-344: Why did you say so? What is the evidence? What are the findings that support these views? There is no/weak linkage between your results and discussion. You need to specify which finding leads to what implication(s).

Line 353: What do you mean “equal number of associations”? What you do mean “the next one”?

Line 354: What do you mean “equal number of waste associations”?

Line 357: No texts are referring to Table 1.

Line 359: “Source: Senekane (2021).” should be put at the table title.

Line 360-381: The logics here are rather weak here. The studies that the researcher of this study cited actually monitored / measured the levels of certain heavy metals in certain media / location / materials. However, it does not necessarily or equally mean that there is linkage between those heavy metals and those media / location / materials.

Line 383-398: The research and/or knowledge gaps should be introduced right after Introduction. The knowledge gaps mentioned in this section, however, are not well explained as no relevant / related literatures are given.

Reviewer 3 Report

Indigenous systems and practices of solid waste management will never disappear because, for indigenous communities, the practice is culturally accepted, people can manage it, and people do not have the support of municipal authorities for waste collection. ISPs are relevant for conservation and the sustainable use of natural resources; however, basic education seems to be a can of worms. Therefore, the problem in the study areas is not ISPs of SWM, per se, but a lack of education and training to community members by authorities in Lesotho on how to engage positively in systems and practices of SWM. The environment can simultaneously be conserved and upheld through indigenous methods of waste management.

This is valuable paper and should be published after minor revision as below.

General Remarks

-          The goal of research should be clearly formulated and placed at the end of the Introduction

-          Abstract should be supplemented with the most important research results

-          Conclusions should include a discussion of the research results

Please do not use abbreviation in Abstract or provide the full name.

Detailed remarks

1.    Abstract – Two last sentences should be removed from Abstract and added to the point 2 “Methods”

2.    Abstract – STATCON, NHREC full names of these abbreviation should be added

3.    Lines 164 – 169 – Some characteristic of analyzed solid waste and their quantities should be added. It would be enriching the study.

4.    Lines 421 – 440. I propose to remove elements name from Abbreviations.

5.    Figure 1, 2, 4, 5, 6 should be self-explaining. Please add proper information.

6.    Table 1 should be self-explaining. Please add proper information.